# Marrying Causal Representation Learning with Dynamical Systems for Science

**Dingling Yao**, **Caroline Muller**, and **Francesco Locatello**

Institute of Science and Technology Austria

## Abstract

Causal representation learning promises to extend causal models to hidden causal variables from raw entangled measurements. However, most progress has focused on proving identifiability results in different settings, and we are not aware of any successful real-world application. At the same time, the field of dynamical systems benefited from deep learning and scaled to countless applications but does not allow parameter identification. In this paper, we draw a clear connection between the two and their key assumptions, allowing us to apply identifiable methods developed in causal representation learning to dynamical systems. At the same time, we can leverage scalable differentiable solvers developed for differential equations to build models that are both identifiable and practical. Overall, we learn explicitly controllable models that isolate the trajectory-specific parameters for further downstream tasks such as out-of-distribution classification or treatment effect estimation. We experiment with a wind simulator with partially known factors of variation. We also apply the resulting model to real-world climate data and successfully answer downstream causal questions in line with existing literature on climate change. Code is available at https://github.com/CausalLearningAI/crl-dynamical-systems.

## 1   Introduction

Causal representation learning (CRL) [54] focuses on *provably* retrieving high-level latent variables from low-level data. Recently, there have been many casual representation learning works compiling, in various settings, different theoretical identifiability results for these latent variables [8, 26, 32, 36, 37, 58, 60, 62, 65, 66, 70, 73]. The main open challenge that remains for this line of work is the broad applicability to real-world data. Following earlier works in disentangled representations (see [38] for a summary of data sets), existing approaches have largely focused on visual data . This is challenging for various reasons. Most notably, it is unclear what the causal variables should be in computer vision problems and what would be interesting or relevant causal questions. The current standard is to test algorithms on synthetic data sets with "made-up" latent causal graphs, e.g., with the object class of a rendered 3d shape causing its position, hue, and rotation [65].

In parallel, the field of machine learning for science [44, 49] shows promising results on various real-world time series data collected from some underlying dynamical systems. Some of these works primarily focus on time-series forecasting, i.e., building a neural emulator that mimics the behavior of the given times series data [12, 13, 25]; while others try to additionally learn an explicit ordinary differential equation simultaneously [9, 10, 15, 18, 23, 56]. However, to the best of our knowledge, none of these methods provide explicit identifiability analysis indicating whether the discovered equation recovers the ground truth underlying governing process given time series observations; or even whether the learned representation relates to the underlying steering parameters. At the same time, many scientific questions are inherently causal, in the sense that physical laws govern the measurements of all the natural data we can record, e.g., across different environments and experimental settings. Identifying such an underlying physical process can boost scientific understanding and reasoning in numerous fields; for example, in climate science, one could conduct

sensitivity analysis of *layer thickness* parameter on atmosphere motion more efficiently, given a neural emulator that identifies the *layer thickness* in its latent space. However, whether mechanistic models can be practically identified from data is so far unclear [54, Table 1].

This paper aims to identify the underlying *time-invariant* physical parameters from real-world time series, such as the previously mentioned *layer thickness* parameter, while still preserving the ability to forecast efficiently. Thus, we connect the two seemingly faraway communities, causal representation learning and machine learning for dynamical systems, by phrasing parameter estimation problems in dynamical systems as a latent variable identification problem in CRL. The benefits are two folds: (1) we can import all identifiability theories for free from causal representation learning works, extending discovery methods with additional identifiability analysis and, e.g., multiview training constructs; (2) we showcase that the scalable mechanistic neural networks [47] recently developed for dynamical systems can be directly employed with causal representation learning, thus providing a scalable implementation for both identifying and forecasting real-world dynamical systems.

Starting by comparing the common assumptions in the field of parameter estimation in dynamical systems and causal representation learning, we carefully justify our proposal to translate any parameter estimation problem into a latent variable identification problem; we differentiate three types of identifiability: *full identifiability*, *partial identifiability* and *non-identifiability*. We describe concrete scenarios in dynamical systems where each kind of identifiability can be theoretically guaranteed and restate *exemplary* identifiability theorems from the causal representation learning literature with slight adaptation towards the dynamical system setup. We provide a step-by-step recipe for reformulating a parameter estimation problem into a causal representation learning problem and discuss the challenges and pitfalls in practice. Lastly, we successfully evaluate our parameter identification framework on various *simulated* and *real-world* climate data. We highlight the following contributions:

- We establish the connection between causal representation learning and parameter estimation for differential equations by pinpointing the alignment of common assumptions between two communities and providing hands-on guidance on how to rephrase the parameter estimation problem as a latent variable identification problem in causal representation learning.

- We equip discovery methods with provably identifiable parameter estimation approaches from the causal representation learning literature and their specific training constructs. This enables us to maintain both the theoretical results from the latter and the scalability of the former.

- We successfully apply causal representation learning approaches to simulated and real-world climate data, demonstrating identifiability via domain-specific downstream causal tasks (OOD classification and treatment-effect estimation), pushing one step further on the applicability of causal representation for real-world problems.

**Remark on the novelty of the paper:** Our main contribution is establishing a connection between the dynamical systems and causal representation learning fields. As such, we do not introduce a new method per se. Meanwhile, this connection allows us to introduce CRL training constructs in methods that otherwise would not have any identification guarantees. Further, it provides the first avenue for causal representation learning applications on real-world data. These are both major challenges in the respective communities, and we hope this paper will serve as a building block for cross-pollination.

## 2 Parameter Estimation in Dynamical Systems

We consider dynamical systems in the form of

$$\dot{\mathbf{x}}(t) = \mathbf{f}_{\boldsymbol{\theta}}(\mathbf{x}(t)) \qquad \mathbf{x}(0) = \mathbf{x}_0, \ \boldsymbol{\theta} \sim p_{\boldsymbol{\theta}}, \ t \in [0, t_{\max}] \tag{1}$$

where $\mathbf{x}(t) \in \mathcal{X} \subseteq \mathbb{R}^d$ denotes the state of a system at time $t$, $f_{\boldsymbol{\theta}} \in \mathcal{C}^1(\mathcal{X}, \mathcal{X})$ is some smooth differentiable vector field representing the constraints that define the system's evolution, characterized by a set of physical parameters $\boldsymbol{\theta} \in \boldsymbol{\Theta} = \boldsymbol{\Theta}_1 \times \cdots \times \boldsymbol{\Theta}_N$, where $\boldsymbol{\Theta} \subseteq \mathbb{R}^N$ is an open, simply connected real space associated with the probability density $p_{\boldsymbol{\theta}}$. Formally, $f_{\boldsymbol{\theta}}$ can be considered as a functional mapped from $\boldsymbol{\theta}$ through $M : \boldsymbol{\Theta} \to \mathcal{C}^1(\mathcal{X}, \mathcal{X})$. In our setup, we consider *time-invariant*, *trajectory-specific* parameters $\boldsymbol{\theta}$ that remain constant for the whole time span $[0, t_{\max}]$, but variable for different trajectories. For instance, consider a robot arm interacting with multiple objects of different mass; a parameter $\boldsymbol{\theta}$ could be the object's masses $m \in \mathbb{R}_+$ in Newton's second law $\ddot{x}(t) = \mathcal{F}(t)/m$, with $\mathcal{F}(t)$ denote the force applied at time $t$. Depending on the object the robot arm interacts with, $m$ can take different values, following the prior distribution $p_{\boldsymbol{\theta}}$. $\mathbf{x}(0) = \mathbf{x}_0 \in \mathcal{X}$

denotes the initial value of the system. Note that higher-order ordinary differential equations can always be rephrased as a first-order ODE. For example, a $\nu$-th order ODE in the following form:

$$x^{(\nu)}(t) = f = (x(t), x^{(1)}(t), \ldots, x^{(\nu-1)}(t), \boldsymbol{\theta}),$$

can be written as $\dot{\mathbf{x}}(t) = f_{\boldsymbol{\theta}}(\mathbf{x}(t))$, where $\mathbf{x}(t) = (x(t), x^{(1)}(t), \ldots, x^{(\nu-1)}(t)) \in \mathbb{R}^{\nu \cdot d}$ denotes state vector constructed by concatenating the derivatives. Formally, the solution of such a dynamical system can be obtained by integrating the vector field over time: $\mathbf{x}(t) = \int_0^t f(\mathbf{x}(\tau), \boldsymbol{\theta}) d\tau$.

> **What do we mean by "parameters"?** The parameters $\boldsymbol{\theta}$ that we consider can be both explicit and implicit. When the functional form of the ODE is given, like Newton's second law, the set of parameters is defined explicitly and uniquely. For real-world physical processes where the functional form of the state evolution is unknown, such as the sea-surface temperature change, we can consider *latitude-related* features as parameters. Overall, we use *parameters* to generally refer to any *time-invariant*, *trajectory-specific* components of the underlying dynamical system.

**Assumption 2.1** (Existence and uniqueness). For every $\mathbf{x}_0 \in \mathcal{X}$, $\boldsymbol{\theta} \in \boldsymbol{\Theta}$, there exists a unique continuous solution $\mathbf{x}_{\boldsymbol{\theta}} : [0, t_{\max}] \to \mathcal{X}$ satisfying the ODE (eq. (1)) for all $t \in [0, t_{\max}]$ [22, 35].

**Assumption 2.2** (Structural identifiability). An ODE (eq. (1)) is *structurally* identifiable in the sense that for any $\boldsymbol{\theta}_1, \boldsymbol{\theta}_2 \in \boldsymbol{\Theta}$, $\mathbf{x}_{\boldsymbol{\theta}_1}(t) = \mathbf{x}_{\boldsymbol{\theta}_2}(t) \, \forall t \in [0, t_{\max}]$ holds if and only if $\boldsymbol{\theta}_1 = \boldsymbol{\theta}_2$ [7, 67, 69].

*Remark* 2.1. Asm. 2.2 implies that it is *in principle* possible to identify the parameter $\boldsymbol{\theta}$ from a trajectory $\mathbf{x}_{\boldsymbol{\theta}}$ [43]. Since this work focuses on providing concrete algorithms that guarantee parameter identifiability *given infinite number of samples*, the structural identifiability assumption is essential as a theoretical ground for further algorithmic analysis. It is noteworthy that a non-structurally identifiable system can become identifiable by reparamatization. For example, linear ODE $\dot{\mathbf{x}}(t) = ab\mathbf{x}(t)$ with parameters $a, b \in \mathbb{R}^2$ is structurally non-identifiable as $a, b$ are commutative. But if we define $c := ab$ as the overall growth rate of the linear system, then $c$ is structurally identifiable.

> **Problem setting.** Given an observed trajectory $\mathbf{x} := (\mathbf{x}_{\boldsymbol{\theta}}(t_0), \ldots, \mathbf{x}_{\boldsymbol{\theta}}(t_T)) \in \mathcal{X}^T$ over the discretized time grid $\mathcal{T} := (t_0, \ldots, t_T)$, our goal is to investigate the identifiability of structurally identifiable parameters by formulating concrete conditions under which the parameter $\boldsymbol{\theta}$ is (i) fully identifiable, (ii) partially identifiable, or (iii) non-identifiable *from the observational data*. We establish the identifiability theory for dynamical systems by converting classical parameter estimation problems [7] into a latent variable identification problem in causal representation learning [54]. For both (i) and (ii), we empirically showcase that existing CRL algorithms with slight adaptation can successfully (*partially*) identify the underlying physical parameters.

## 3 Identifiability of Dynamical Systems

This section provides different types of theoretical statements on the identifiability of the underlying *time-invariant*, *trajectory-specific* physical parameters $\boldsymbol{\theta}$, depending on whether the functional form of $f_{\boldsymbol{\theta}}$ is known or not. We show that the parameters from an ODE with a known functional form can be *fully identified* while parameters from unknown ODEs are in general *non-identifiable*. However, by incorporating some weak form of supervision, such as multiple similar trajectories generated from certain overlapping parameters [16, 39, 65, 71], parameters from an unknown ODE can also be *partially identified*. Detailed proofs of the theoretical statements are provided in App. B.

### 3.1 Identifiability of Dynamical Systems with Known Functional Form

We begin with the identifiability analysis of the physical parameters of an ODE with **known** functional form. Many real-world data we record are governed by known physical laws. For example, the bacteria growth in microbiology could be modeled with a simple logistic equation under certain conditions, where the parameter of interest in this case would be the *growth rate* $r \in \mathbb{R}_+$ and *maximum capacity* $K \in \mathbb{R}_+$. Identifying such parameters would be helpful for downstream analysis. To this end, we introduce the definition of *full identifiability* of a physical parameter vector $\boldsymbol{\theta}$.

**Definition 3.1** (Full identifiability). A parameter vector $\boldsymbol{\theta} \in \boldsymbol{\Theta}$ is fully identified if the estimator $\hat{\boldsymbol{\theta}}$ converges to the ground truth parameter $\boldsymbol{\theta}$ almost surely.

**Definition 3.2** (ODE solver). An ODE solver $F : \boldsymbol{\Theta} \to \mathcal{X}^T$ computes the solution $\mathbf{x}$ of the ODE $f_{\boldsymbol{\theta}} = M(\boldsymbol{\theta})$ (eq. (1)) over a discrete time grid $T = (t_1, \ldots, t_T)$.

Table 1: **Comparing typical assumptions** of parameter estimation for dynamical systems and latent variable identification in causal representation learning. We justify that the common assumptions in both fields are aligned, providing theoretical ground for applying identifiable CRL methods to learning-based parameter estimation approaches in dynamical systems.

| param. estimation | | CRL | | Explanation |
|---|---|---|---|---|
| *ref* | *assumption* | *assumption* | *ref* | |
| 2.1 | *existence & uniqueness* ○——○ | *determ. gen.* | 3.1 | Both 2.1 and 3.1 implies deterministic generative process. |
| | | ○ $supp(\boldsymbol{\theta}) = \boldsymbol{\Theta}$ | 3.3 | 2.1 implies 3.3 as $\mathbf{x}_{\boldsymbol{\theta}}$ uniquely exists for all $\boldsymbol{\theta} \in \boldsymbol{\Theta}$. |
| 2.2 | *structural identifiability* ○—— | ——○ *injectivity* | 3.2 | 2.2 implies 3.2 of the solution $\mathbf{x}_{\boldsymbol{\theta}}$. |

**Corollary 3.1** (Full identifiability with known functional form). *Consider a trajectory* $\mathbf{x} \in \mathcal{X}^T$ *generated from a ODE* $f_{\boldsymbol{\theta}}(\mathbf{x}(t))$ *satisfying Asms. 2.1 and 2.2, let* $\hat{\boldsymbol{\theta}}$ *be an estimator minimizing the following objective:*

$$\mathcal{L}(\hat{\boldsymbol{\theta}}) = \left\| F(\hat{\boldsymbol{\theta}}) - \mathbf{x} \right\|_2^2 \tag{2}$$

*then the parameter* $\boldsymbol{\theta}$ *is **fully-identified** (Defn. 3.1) by the estimator* $\hat{\boldsymbol{\theta}}$.

*Remark* 3.1. The estimator $\hat{\theta}$ of eq. (2) is considered as some learnable parameters that can be directly optimized. If we have multiple trajectories $\mathbf{x}$ generated from different realizations of $\boldsymbol{\theta} \sim p_{\boldsymbol{\theta}}$, we can also amortize the prediction $\hat{\boldsymbol{\theta}}$ using a smooth encoder $g : \mathcal{X}^T \to \boldsymbol{\Theta}$. In this case, the loss above can be rewritten as: $\mathcal{L}(g) = \mathbb{E}_{\mathbf{x},t}[\|F(g(\mathbf{x})) - \mathbf{x}(t)\|_2^2]$, then the optimal encoder $g^* \in \text{argmin} \, \mathcal{L}(g)$ can generalize to unseen trajectories $\mathbf{x}$ that follow the same class of physical law $f$ and fully identify their trajectory-specific parameters $\boldsymbol{\theta}$.

**Discussion.** Many works on machine learning for dynamical system identification follow the principle presented in Cor. 3.1, and most of them solely differ concerning the architecture they choose for the ODE solver. For example, SINDy-like ODE discovery methods [9, 10, 23, 24, 47] approximate the ground truth vector field $f$ using a linear weighted sum over a set of library functions and learn the linear coefficients by sparse regression. For any ODE $f$ that is linear in $\boldsymbol{\theta}$, i.e., the ground truth vector field is in the form of $f_{\boldsymbol{\theta}}(\mathbf{x}, t) = \sum_{i=1}^m \theta_i \phi_i(\mathbf{x})$ for a set of known base functions $\{\phi_i\}_{i \in [m]}$, SINDy-like approaches can fully identify the parameters by imposing some sparsity constraint. Another line of work, gradient matching [68], estimates the parameters probabilistically by modeling the vector field $f_{\boldsymbol{\theta}}$ using a Gaussian Process (GP). The modeled solution $\mathbf{x}(t)$ is thus also a GP since GP is closed under integrals (a linear operator). Given the functional form of $f_{\boldsymbol{\theta}}$, the model aims to match the estimated gradient $\dot{\mathbf{x}}$ and the evaluated vector field $f_{\boldsymbol{\theta}}(\mathbf{x}(t))$ by maximizing the likelihood, which is equivalent to minimizing the least-squares loss (eq. (2)) under Gaussianity assumptions. Hence, the gradient matching approaches can *theoretically* identify the underlying parameters under Cor. 3.1. Formal statements and proofs for both SINDy-like and gradient matching approaches are provided in App. B. *Note that most ODE discovery approaches [9, 10, 23, 24, 47, 68] refrain from making identifiability statements and explicitly states it is unknown which settings yield identifiability.*

## 3.2 Identifiability of Dynamical Systems without Known Functional Form

In traditional dynamical systems, identifiability analysis usually assumes the functional form of the ODE is known [43]; however, for most real-world time series data, the functional form of underlying physical laws remains uncovered. Machine learning-based approaches for dynamical systems work in a black-box manner and can clone the behavior of an unknown system [12, 13, 46], but understanding and identifiability guarantees of the learned parameters are so far missing. Since most of the physical processes are inherently steered by a few underlying *time-invaraint* parameters, identifying these parameters can be helpful in answering downstream scientific questions. For example, identifying climate zone-related parameters from sea surface temperature data could improve understanding of climate change because the impact of climate change significantly differs in polar and tropical regions. Hence, we aim to provide identifiability analysis for the underlying parameters of an unknown dynamical system by converting the classical parameter estimation problem of dynamical systems into a latent variable identification problem in causal representation learning. We start by listing the common assumptions in CRL and comparing the ground assumptions between these two fields.

**Assumption 3.1** (Determinism). The data generation process is deterministic in the sense that observation $\mathbf{x}$ is generated from some latent vector $\boldsymbol{\theta}$ using a deterministic solver $F$ (Defn. 3.2).

**Assumption 3.2** (Injectivity). For each observation $\mathbf{x}$, there is only one corresponding latent vector $\boldsymbol{\theta}$, i.e., the ODE solve function $F$ (Defn. 3.2) is injective in $\boldsymbol{\theta}$.

**Assumption 3.3** (Continuity and full support). $p_{\boldsymbol{\theta}}$ is smooth and continuous on $\boldsymbol{\Theta}$ with $p_{\boldsymbol{\theta}} > 0$ a.e.

**Assumption justification.** Tab. 1 summarizes common assumptions in traditional parameter estimation in dynamical systems and causal representation learning literature. We observe strong alignment between the ground assumptions in these two fields that justifies our idea of employing causal representation learning methods in parameter estimation problems for dynamical systems: (1) Asm. 2.1 implies that given a fixed initial value $\mathbf{x}_0 \in \mathcal{X}$, there exists a unique solution $\mathbf{x}(t)$, $t \in [0, t_{\max}]$ for any $f_{\boldsymbol{\theta}}$ with $\boldsymbol{\theta} \in \boldsymbol{\Theta}$. In other words, parameter domain $\boldsymbol{\Theta}$ is fully supported (Asm. 3.3), and these ODE solving processes from $F(\boldsymbol{\theta})$ (Defn. 3.2) are deterministic, which aligns with the standard Asm. 3.1 in CRL. Since the ODE solution $F(\boldsymbol{\theta})$ (§ 2) is continuous by definition, the continuity assumption from CRL (Asm. 3.3) is also fulfilled. (2) Asm. 2.2 emphasizes that each trajectory $\mathbf{x}$ can only be uniquely generated from one parameter vector $\boldsymbol{\theta} \in \boldsymbol{\Theta}$, which means the generating process $F$ (Defn. 3.2) is injective in $\boldsymbol{\theta}$ (Asm. 3.2).

Next, we reformulate the parameter estimation problem in the language of causal representation learning. We first cast the generative process of the dynamical system $f_{\boldsymbol{\theta}}(\mathbf{x}(t))$ as a latent variable model by considering the underlying physical parameters $\boldsymbol{\theta} \sim p_{\boldsymbol{\theta}}$ as a set of *latent variables*. Given a trajectory $\mathbf{x}$ generated by a set of underlying factors $\boldsymbol{\theta}$ based on the vector field $f_{\boldsymbol{\theta}}(\mathbf{x}(t))$, we consider the observed trajectory as some *unknown nonlinear* mixing of the underlying $\boldsymbol{\theta}$, with the mixing process specified by individual vector field $f_{\boldsymbol{\theta}}(\mathbf{x}(t))$. This interpretation of observations aligns with the standard setup of causal representation learning; for instance, high-dimensional images are usually generated from some lower-dimensional latent generating factors through an unknown nonlinear process. Thus, estimating the parameters of unknown dynamical systems becomes equivalent to inferring the underlying generating factors in causal representation learning.

After transforming the parameter estimation into a latent variable identification problem in CRL, we can directly invoke the identifiability theory from the literature. Based on Locatello et al. [38, Theorem 1.], we conclude that the underlying parameters from an unknown system are in general ***non-identifiable***. Nevertheless, several works proposed different weakly supervised learning strategies that can *partially identify* the latent variables [2, 8, 16, 39, 65, 71]. To this end, we define partial identifiability in the context of dynamical systems by slightly adapting the definition of block-identifiability proposed by Von Kügelgen et al. [65]:

**Definition 3.3** (Partial identifiability). A partition $\boldsymbol{\theta}_S := (\boldsymbol{\theta}_i)_{i \in S}$ with $S \subseteq [N]$ of parameter $\boldsymbol{\theta} \in \boldsymbol{\Theta}$ is partially identified by an encoder $g : \mathcal{X}^T \to \boldsymbol{\Theta}$ if the estimator $\hat{\boldsymbol{\theta}}_S := g(\mathbf{x})_S$ contains all and only information about the ground truth partition $\boldsymbol{\theta}_S$, i.e. $\hat{\boldsymbol{\theta}}_S = h(\boldsymbol{\theta}_S)$ for some invertible mapping $h : \boldsymbol{\Theta}_S \to \boldsymbol{\Theta}_S$ where $\boldsymbol{\Theta}_S := \times_{i \in S} \boldsymbol{\Theta}_i$.

Note that the inferred partition $\hat{\boldsymbol{\theta}}_S$ can be a set of *entangled* latent variables rather than a single one. In the multivariate case, one can consider the $\hat{\boldsymbol{\theta}}_S$ as a bijective mixture of the ground truth parameter $\boldsymbol{\theta}_S$.

**Corollary 3.2** (Identifiability without known functional form). *Assume a dynamical system $f$ satisfying Asms. 2.1 and 2.2, a pair of trajectories $\mathbf{x}, \tilde{\mathbf{x}}$ generated from the same system $f$ but specified by different parameters $\boldsymbol{\theta}, \tilde{\boldsymbol{\theta}}$, respectively. Assume a partition of parameters $\boldsymbol{\theta}_S$ with $S \subseteq [N]$ is shared across the pair of parameters $\boldsymbol{\theta}, \tilde{\boldsymbol{\theta}}$. Let $g : \mathcal{X}^T \to \boldsymbol{\Theta}$ be some smooth encoder and $\hat{F} : \boldsymbol{\Theta} \to \mathcal{X}^T$ be some left-invertible smooth solver that minimizes the following objective:*

$$\mathcal{L}(g, \hat{F}) = \mathbb{E}_{\mathbf{x}, \tilde{\mathbf{x}}} \underbrace{\|g(\mathbf{x})_S - g(\tilde{\mathbf{x}})_S\|_2^2}_{\text{Alignment}} + \underbrace{\left\|\hat{F}(g(\mathbf{x})) - \mathbf{x}\right\|_2^2 + \left\|\hat{F}(g(\tilde{\mathbf{x}})) - \tilde{\mathbf{x}}\right\|_2^2}_{\text{Sufficiency}}, \qquad (3)$$

*then the shared partition $\boldsymbol{\theta}_S$ is partially identified (Defn. 3.3) by $g$ in the statistical setting.*

**Discussion.** We remark that an implicit ODE solver $\hat{F}$ is introduced in eq. (3) because the functional form $f_{\boldsymbol{\theta}}$ is unknown. Intuitively, Cor. 3.2 provides partial identifiability results for the shared partition of parameters between two trajectories. We can consider the trajectories to be different simulation experiments but with certain sharing conditions, such as two wind simulations that share the same *layer thickness* parameter. This partial identifiability statement is mainly concluded from the theory in the multiview CRL literature [2, 8, 16, 39, 54, 65, 71]. Note that this corollary is *one exemplary*

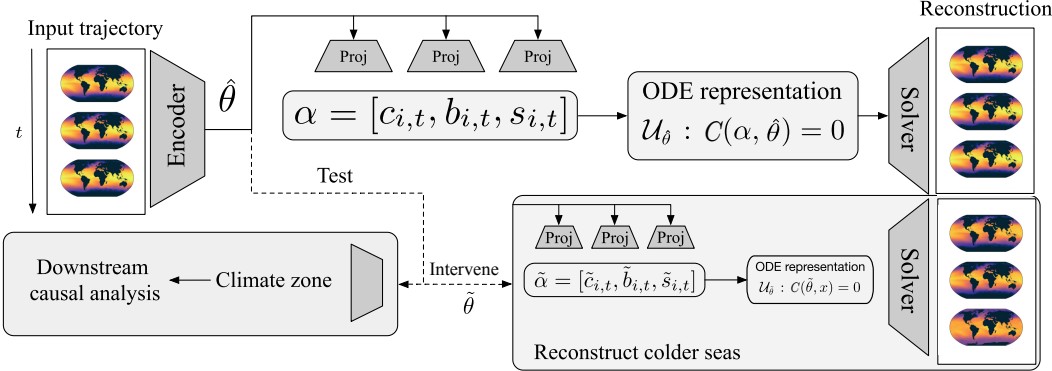

Figure 1: **Model overview with sea surface temperature inputs:** Our *mechanistic identifier* extracts the underlying time-invariant latitude-related parameters $\boldsymbol{\theta}$, providing a versatile neural emulator for downstream causal analysis.

*demonstration* of achieving partial identifiability in dynamical systems. Many identifiability results from the causal representation works can be reformulated similarly by replacing their decoder with a differentiable ODE solver $\hat{F}$. The high-level idea of multiview CRL is to identify the shared part between different views by enforcing alignment on the shared coordinates while preserving a sufficient information representation. *Alignment* can be obtained by either minimizing the $L_2$ loss between the encoding from different views on the shared coordinates [16, 65, 71] or maximizing the correlation on the shared dimensions correspondingly [41, 42]; *Sufficiency* of the learned representation is often prompted by maximizing the entropy [16, 65, 71, 74] or minimizing the reconstruction error [2, 8, 39, 54]. Other types of causal representation learning works will be further discussed in § 5.

## 4    CRL-construct of Identifiable Neural Emulators for Dynamical Systems

This section provides a step-by-step construct of a neural emulator that can (1) identify the *time-invariant, trajectory-specific* physical parameters from some unknown dynamical systems if the identifiability conditions are met and (2) efficiently forecast future time steps. Identifiability can be guaranteed by employing causal representation learning approaches (§ 3) while forecasting ability can be obtained by using an efficient mechanistic solver [47] as a decoder. For the sake of simplicity, we term these identifiable neural emulators as *identifiers*. We remark that the general architecture remains consistent for most CRL approaches, while the learning object differs slightly in *latent regularization*, which is specified by individual identifiability algorithms. Intuitively, the *latent regularization* can be interpreted as an additional constraint put on the learned encodings imposed by the setting-specific assumptions, such as the *alignment* term in multiview CRL (Cor. 3.2). In the following, we demonstrate building an *identifier* in the multiview setting from scratch and showcase how it can be easily generalized to other CRL approaches with slight adaptation.

**Architecture.** Since the parameters of interest are *time-invariant* and *trajectory-specific* (§ 2), we input the whole trajectory $\mathbf{x} = (\mathbf{x}(t_1), \dots, \mathbf{x}(t_T))$ to a smooth encoder $g : \mathcal{X}^T \to \boldsymbol{\Theta}$, as shown in Fig. 1. Then, we decode the trajectory $\hat{\mathbf{x}}$ from estimated parameter vector $\hat{\boldsymbol{\theta}} := g(\mathbf{x})$ using a mechanistic solver [47]. The high-level idea of mechanistic neural networks is to approximate the underlying dynamical system using a set of explicit ODEs $\mathcal{U}_{\hat{\boldsymbol{\theta}}} : C(\boldsymbol{\alpha}, \hat{\boldsymbol{\theta}}) = 0$ with learnable coefficients $\boldsymbol{\alpha} \in \mathbb{R}^{d_\alpha}$. The explicit ODE family $\mathcal{U}_{\hat{\boldsymbol{\theta}}}$ can then be interpreted as a constrained optimization problem and can thus be solved using a *neural relaxed linear programming solver* [47, Sec 3.1].

In more detail, the original design of MNN predicts the coefficients from the input trajectory $\mathbf{x}$ using an MNN encoder $g_{\mathrm{mnn}}$; however, as we enforce the estimated parameter $\boldsymbol{\theta}$ to preserve *sufficient* information of the entire trajectory $\mathbf{x}$, we instead predict the coefficients $\boldsymbol{\alpha}$ from the estimated parameter $\hat{\boldsymbol{\theta}}$ with the encoder $g_{\mathrm{mnn}} : \boldsymbol{\Theta} \to \mathbb{R}^{d_\alpha}$. Formally, the coefficients $\boldsymbol{\alpha}$ are computed as $\boldsymbol{\alpha} = g_{\mathrm{mnn}}(\hat{\boldsymbol{\theta}})$ where $\hat{\boldsymbol{\theta}} = g(\mathbf{x})$. The resulting ODE family $\mathcal{U}_{\hat{\boldsymbol{\theta}}}$ provides a broad variability of ODE

parametrizations. A detailed formulation of $\mathcal{U}_{\hat{\theta}}$ at $t$ [47, eq. (3)] is given by

$$\underbrace{\sum_{i=0}^{l} c_i(t;\hat{\boldsymbol{\theta}})u^{(i)}}_{\text{linear terms}} + \underbrace{\sum_{j=0}^{r} \phi_k(t;\hat{\boldsymbol{\theta}})g_k(t,\{u^{(j)}\})}_{\text{nonlinear terms}} = b(t;\hat{\boldsymbol{\theta}}), \tag{4}$$

where $u^{(i)}$ is $i$-th order approximations of the ground truth state $\mathbf{x}$. Like in any ODE solving in practice, solving eq. (4) requires discretization of the continuous coefficients in time (e.g., $c_i(t;\hat{\boldsymbol{\theta}})$). Discretizing the ODE representation $\mathcal{U}_{\hat{\theta}}$ gives rise to:

$$\sum_{i=0}^{l} c_{i,t}u_t^{(i)} + \sum_{j=0}^{r} \phi_{k,t}g_k(\{u_t^{(j)}\}) = b_t \quad s.t. \quad (u_{t_1}, u'_{t_1}, \dots) = \omega, \tag{5}$$

where $\omega$ denotes the initial state vector of the ODE representation $\mathcal{U}_{\hat{\theta}}$. To this end, we present the explicit definition of the learnable coefficients $\boldsymbol{\alpha} := (c_{i,t}, \phi_{k,t}, b_t, s_t, \omega)$ with $t \in \mathcal{T}, i \in [l], k \in [r]$, which is a concatenation of linear coefficients $c_{i,t}$, nonlinear coefficients $\phi_{i,k}$, adaptive step sizes $s_t$ and initial values $\omega$. Note that we dropped the $\hat{\boldsymbol{\theta}}$ in the notation for simplicity, but all of these coefficients $\boldsymbol{\alpha}$ are predicted from $\hat{\boldsymbol{\theta}}$, as described previously. At last, MNN converts ODE solving into a constrained optimization problem by representing the $\mathcal{U}_{\hat{\theta}}$ using a set of constraints, including ODE equation constraints, initial value constraints, and smoothness constraints [47, Sec 3.1.1]. This optimization problem is then solved by *neural relaxed linear programming* solver [47, Sec 3.1] in a time-parallel fashion, thus making the overall mechanistic solver scalable and GPU-friendly.

**Learning objective and latent regularizers.** Depending on whether the functional form of the underlying dynamical system is known or not, the proposed neural emulator can be trained using the losses given in Cor. 3.1 or Cor. 3.2, respectively. When the functional form is unknown, we employ CRL approaches to *partially* identify the physical parameters. We remark that the causal representation learning schemes mainly differ in the latent regularizers, specified by the assumptions and settings. Therefore, we provide a more extensive summary of different causal representation learning approaches and their corresponding latent regularizer in Tab. 6.

## 5 Related Work

**Multi-environment CRL.** Another important line of work in causal representation learning focuses on the multi-environment setup, where the data are collected from multiple different environments and thus *non-identically distributed*. Causal variable identifiability are shown under *single node intervention per node* with parametric assumptions on the mixing functions [3, 58, 62, 73] or on the latent causal model [11, 58]. These parametric assumptions can be lifted by additionally assuming *paired intervention per node*, as demonstrated by [63, 66]. Overall, given the fruitful literature in multi-environment causal representation learning, we believe applying multi-environments methods to build identifiable neural emulators (§ 4) would be an exciting future avenue.

**CRL and dynamical systems.** Recent CRL works have been tackling the parameter identification problem in dynamical systems in a parametric setting. For instance, Rajendran et al. [50] considers a Gaussian linear time invariant system with control input and Balsells-Rodas et al. [5] assumes a switching dynamical system. By contrast, we show identifiability in a more general setting without specific parameteric assumptions on the dynamical systems and prove different granularity of parameter identification under different system prior system knowledge (full identifiability if parametric form is known (Cor. 3.1) and partial identifiability when the system is unknown (Cor. 3.2)). Another closely related line of works, temporal causal representation learning, typically assume an *"intervenable"* time series in the *latent space*, splitting the latent variables into two partitions, with one following the default dynamics and the other following the intervened dynamics. The goal of temporal CRL is to provably retrieve these *time-varying* latent causal variables, such as inferring the position of a ball from raw images over time [27, 30, 33, 36, 37, 72]. Unlike these temporal CRL works, our approach models dynamics directly in the observational space, focusing on the *time-invariant, trajectory-specific* physical parameters such as gravity or mass. Overall, our framework addresses a different hierarchy of problems. We believe both problems are orthogonal yet equally important, encouraging cross-pollination in future work.

**ODE discovery.** The ultimate goal of ODE discovery is to learn a human-interpretable equation for an unknown system, given discretized observations generated from this system. Recently, many machine learning frameworks have been used for ODE discovery, such as sparse linear regression [9, 10, 23, 53], symbolic regression [6, 15, 18], simulation-based inference [14, 56].

Becker et al. [6], d'Ascoli et al. [18] exploit transformer-based approaches to dynamical symbolic regression for univariate ODEs, which is extended by d'Ascoli et al. [15] to multivariate case. Schröder and Macke [56] employs *simulation-based variational inference* to jointly learn the operators (like addition or multiplication) and the coefficients. However, this approach typically runs simulations inside the training loop, which could introduce a tremendous computational bottleneck when the simulator is inefficient. On the contrary, our approach works offline with pre-collected data, avoiding simulating on the fly. Although ODE discovery methods can provide symbolic equations for data from an unknown trajectory, the inferred equation does not have to align with the ground truth. In other words, theoretical identifiability guarantees for these methods are still missing.

**Identifiability of dynamical systems.** Identifiability of dynamical systems has been studied on a *case-by-case* basis in traditional system identification literature [4, 43, 64]. Liang and Wu [34] studied ODE identifiability under measurement error. Scholl et al. [55] investigated the identifiability of ODE discovery with non-parametric assumption, but only for univariate cases. More recently, several works have advanced in identifiability analysis of *linear* ODEs from a *single* trajectory [17, 48, 59]. Overall, current theoretical results cannot conclude whether an unknown nonlinear ODE can be identified from observational data. Hence, in our work, we do not aim to identify the whole equation of the dynamical systems but instead focus on identifying the time-invariant parameters.

# 6  Experiments

This section provides experiments and results on both simulated and real-world climate data. We first validate full parameter identifiability (Cor. 3.1) under a wide range of known dynamical systems, as demonstrated in § 6.1. Next, we consider in §§ 6.2 and 6.3 time series data governed by an unknown physical process, so we employ the multiview CRL approach together with mechanistic neural networks to build our identifiable neural emulator (termed as *mechanistic identifier*), following the steps in § 4. We compare *mechanistic identifier* with three baselines: (1) *Ada-GVAE* [39], a traditional multiview model that uses a vanilla decoder instead of a mechanistic solver. (2) *Time-invariant MNN*, proposed by [47]. We choose this variant of MNN as our baseline for a fair comparison. (3) *Contrastive identifier*, a contrastive loss-based CRL approach without a decoder [16, 65, 71]. We train *mechanistic identifier* using eq. (3) and other baselines following the steps given in the original papers. After training, we evaluate these methods on their identifiability and long-term forecasting capability.

## 6.1  Theory Validation: ODEBench

We demonstrate point-wise parameter identification results (presented as RMSE, mean $\pm$ std) over ODE system with known functional forms, including 63 dynamical systems from ODEBench [15] and the Cart-Pole system inspired by [72]. Results are summarized in Tab. 7. For each system, we sample 100 tuples of the parameters $\theta$ within a valid range to e.g., preserve the chaotic properties. For each tuple, we solve the problem as outlined in Cor. 3.1, by either regressing the estimated observation $F(\hat{\boldsymbol{\theta}})$ onto the true observation $\mathbf{x}$, or equivalently, regressing the estimated vector field $f_{\boldsymbol{\theta}}(\mathbf{x})$ onto the derivatives $\dot{\mathbf{x}}$, given the same initial conditions. Note the data derivatives $\dot{\mathbf{x}}$ can be obtained from numerical approximation in case it is not given a priori. The resulting root-mean-square deviation (RMSE) is calculated and averaged across the parameter dimensions. We report the mean and standard deviation of these averaged RMSEs over the 100 independent runs. From Tab. 7, we observe highly accurate point estimates for all stationary system parameters $\theta$, thereby validating Cor. 3.1 across various experimental settings.

## 6.2  Wind Simulation

**Experimental setup.** Our experiment considers longitudinal and latitudinal wind velocities (also termed $u, v$ wind components) from the global wind simulation data generated by various *layer-thickness* parameters. Fig. 2 depicts the wind simulation output at a certain time point. To train the multiview approaches, we generate a tuple of three views: After sampling the first view $\mathbf{x}^1$ randomly throughout the whole training set, we sample another trajectory $\mathbf{x}^2$ from a different location which shares the same simulation condition as the first one, compared to the first view, the third view $\mathbf{x}^3$ is then sampled from another simulation but at the same location. Overall, $\mathbf{x}^1, \mathbf{x}^2$ share the global simulation conditions like the *layer thickness* parameter

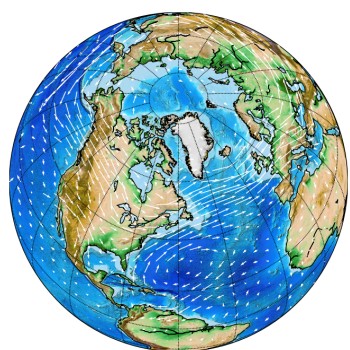

Figure 2: **Wind simulation**: $u, v$ components [m/s] of simulated air motion over the globe.

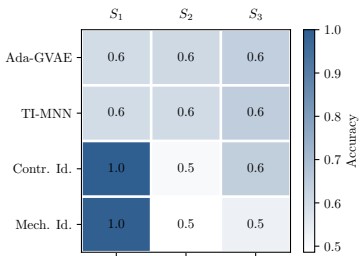

Figure 3: **Prediction accuracy on *layer thickness* parameter on wind simulation data**, evaluated on individual encoding partitions $S_1, S_2, S_3$. Results averaged from three random runs.

Table 2: **Performance evaluation on the SST-V2 data on various types of tasks**. Results averaged over three random seeds with standard deviation, provided as (m ± std).

| | SST V2 | | |
|---|---|---|---|
| | **Acc.(ID)($\uparrow$)** | **Acc.(OOD)($\uparrow$)** | **Forecast. error($\downarrow$)** |
| Ada-GVAE | $0.468 \pm 0.001$ | $0.467 \pm 0.000$ | $0.043 \pm 0.044$ |
| TI-MNN | $0.697 \pm 0.049$ | $0.668 \pm 0.074$ | $0.024 \pm 0.016$ |
| Contr. Identifier | $\mathbf{0.904 \pm 0.011}$ | $\mathbf{0.861 \pm 0.022}$ | ✗ |
| **Mech. Identifier** | $0.902 \pm 0.005$ | $0.824 \pm 0.016$ | $\mathbf{0.007 \pm 0.003}$ |

while $\mathbf{x}^1, \mathbf{x}^3$ only share the local features. All three views share global atmosphere-related features that are not specified as simulation conditions. More details about the data generation process and training pipeline are provided in App. C.2.

**Parameter identification.** In this experiment, we use the learned representation to classify the ground-truth labels generated by discretizing the generating factor *layer thickness*, and report the accuracy in Fig. 3. In more detail, we use `latent dim=12` for all models and split the learned encodings into three partitions $S_1, S_2, S_3$, with four dimensions each. Then, we individually predict the ground truth *layer thickness* labels from each partition. According to the previously mentioned view-generating process, the *layer thickness* parameter should be encoded in $S_1$ for both *contrastive* and *mechanistic identifiers*. This hypothesis is verified by Fig. 3 since both *contrastive* and *mechanistic identifiers* show a high accuracy of `acc≈1` in the first partition $S_1$ and low accuracy in other partitions. On the contrary, *Ada-GVAE* and *TI-MNN* performed significantly worse with an average acc. of 60% everywhere. Overall, Fig. 3 shows both the necessity of explicit time modeling using MNN solver (compared to *Ada-GVAE*) and identifiability power of multiview CRL (compared to *TI-MNN*).

### 6.3 Real-world Sea Surface Temperature

**Experimental setup.** We evaluate the models on sea surface temperature dataset *SST-V2* [21]. For the multiview training, we generate a pair trajectories from a small neighbor region ($\pm 5°$) along the ***same latitude***. We believe these pairs share certain climate properties as the locations from the same latitude share *roughly* the amount of direct sunlight which will directly affect the sea surface temperature. Further infromation about the dataset and training procedure is provided in App. C.2.

**Time series forecasting.** We chunk the time series into slices of 4 years in training while keeping last four years as out-of-distribution forecasting task. To predict the last chunk, we input data from 2015 to 2018 to get the learned representation $\hat{\boldsymbol{\theta}}$. Since we assume $\hat{\boldsymbol{\theta}}$ to be *time-inavriant*, we decode $\hat{\boldsymbol{\theta}}$ together with 10 initial steps of 2019 to predict the last chunk. Note that *contrastive identifier* is excluded from this task as it does not have a decoder. As shown in Tab. 2, the forecasting performance of *mechanistic Identifier* surpasses *Ada-GVAE* by a great margin, showcasing the superiority of integrating scalable mechanistic solvers in real-world time series datasets. At the same time, *TI-MNN* performed worse and unstably despite the MNN component, verifying the need of the additional information bottleneck (parameter encoder $g$) and the multiview learning scheme.

**Climate-zone classification.** Since there is no ground truth latitude-related parameters available, we design a downstream classification task that verifies our learned representation encodes the latitude-related information. The goal of the task is to predict the climate zone *(tropical, temperate, polar)* from the learned *shared* representation because the latitude uniquely defines climate zones. We evaluated the methods in both *in-distribution* (ID) and *out-of-distribution* (OOD) setup for all baselines. In the OOD setting, we input data from longitude $10°$ to longitude $360°$ when training the classifier while keeping the first 10 degree as our out-of-distribution test data. Tab. 2 show that both *contrastive* and *mechanistic identifiers* perform decently, supporting the applicability of identifiable multiview CRL algorithms in dynamical systems. Overall, the performance of multiview CRL-based approaches (*contrastive and mechanistic identifiers*) far exceeds *Ada-GVAE* and *TI-MNN*, again showcasing the superiority of the combination of causal representation learning and mechanistic solvers.

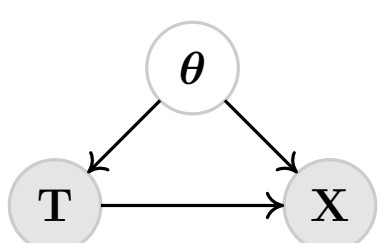
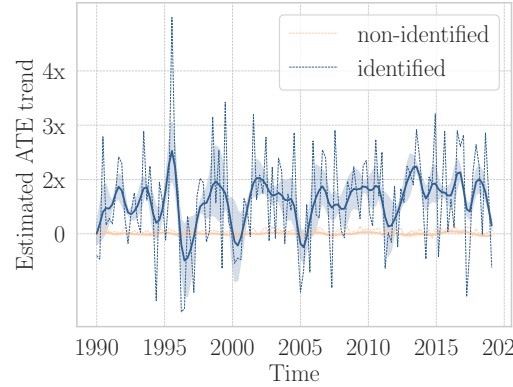

Figure 4: *Left*: Underlying causal model for SST-V2 data, $\boldsymbol{\theta}$: covariates (latitude-related parameters of interest), $\mathbf{X}$: outcome (zonal average temperature), $\mathbf{T}$: treatment (tropical $\mathbf{T} = 0$ or polar $\mathbf{T} = 1$). *Right:* Comparison on ATE change ratio between identified and non-identified parameters, computed by ${ATE(year) - ATE(1990)}/{ATE(1990)}$, averaged over three runs.

**Average treatment effect estimation.** We further investigate the effect of climate zone on average temperature along one specific latitude through *average treatment effect* (ATE) estimation. Formally, we consider the latitudinal average temperature as outcome $Y$, two climate zones (*tropical* $(T = 0)$, *polar*$(T = 1)$) as binary treatments, and the predicted latitude-specific features as unobserved mediators. Formally, ATE is defined as: ATE $:= \mathbb{E}[Y|do(T = 1)] - \mathbb{E}[Y|do(T = 0)]$. Since ATE cannot be computed directly [20], we estimate it using the popular *AIPW* estimator [52]. Fig. 4 illustrates that the estimated ATE from the non-identified representation lacks a discernible pattern [51] whereas the identified representation exhibits a noisy yet clear increasing trend, indicating the global warming effect. This is because the non-identified representation failed to isolate the covariates $\theta$, leading to biased treatment effect estimates. To estimate treatment effects, the covariates (i.e., the latitude-related parameters we identify) must not be influenced by the treatment (i.e., the climate zones). Otherwise, they become confounders, leading to incorrect estimates [19].

# 7 Limitations and Conclusion

In this paper, we build a bridge between causal representation learning and dynamical system identification. By virtue of this connection, we successfully equipped existing mechanistic models (focusing on [47] in practice for scalability reasons) with identification guarantees. Our analysis covers a large number of papers, including [9, 10, 23, 24, 47, 68] explicitly refraining from making identifiability statements. At the same time, our work demonstrated that causal representation learning training constructs are ready to be applied in the real world, and the connection with dynamical systems offers untapped potential due to its relevance in the sciences. This was an overwhelmingly acknowledged limitation of the causal representation learning field [3, 11, 16, 39, 58, 62, 65, 71]. Having clearly demonstrated the mutual benefit of this connection, we hope that future work will scale up identifiable mechanistic models and apply them to even more complex dynamical systems and real scientific questions. Nevertheless, this paper has several technical limitations that could be addressed in future work. First of all, the proposed theory explicitly requires *determinism* as one of the key assumptions (Asm. 3.1), which directly excludes another important type of differential equation: Stochastic Differential Equations. Second, we assume we directly observe the state $\mathbf{x}$ without considering measurement noise. Although the empirical results were promising on real-world noisy data (§ 6.3), we believe explicitly modeling measurement noise would elevate the theory. Finally, our identifiability analysis focuses on the infinite data regime, which is unrealistic in real-world scenarios.

## Acknowledgments

We thank Niklas Boers for recommending the SpeedyWeather simulator and Valentino Maiorca for guidance on Fourier transformation for SST data. We are also grateful to Shimeng Huang and Riccardo Cadei for their feedback on the treatment effect estimation experiment and to Jiale Chen and Adeel Pervez for their assistance with the solver implementation. Finally, we appreciate the anonymous reviewers for their insightful suggestions, which helped improve the manuscript.

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

# Appendix

## Table of Contents

## A  Notation and Terminology

$f$      Vector field

$N$      Dimensionality of parameter $\boldsymbol{\theta}$

$\Theta$      Parameter domain

$\boldsymbol{\theta}$      Time-invariant parameters for ODE $f$

$M$      Function that maps from $\boldsymbol{\theta}$ to $f_{\boldsymbol{\theta}}$

$T$      Number of time steps

$t_{\max}$      End of the time span

$\mathcal{T}$      Discretized time grid of size T

$\mathbf{x}(t)$      ODE solution at time $t$

$\mathbf{x}$      ODE solution for the time grid $\mathcal{T}$

$\mathcal{X}$      State-space domain

$\mathcal{X}^T$      Trajectory domain over time grid $\mathcal{T}$

## B  Proofs

### B.1  Proofs for full identifiability

**Corollary 3.1** (Full identifiability with known functional form). *Consider a trajectory* $\mathbf{x} \in \mathcal{X}^T$ *generated from a ODE* $f_{\boldsymbol{\theta}}(\mathbf{x}(t))$ *satisfying Asms. 2.1 and 2.2, let* $\hat{\boldsymbol{\theta}}$ *be an estimator minimizing the following objective:*

$$\mathcal{L}(\hat{\boldsymbol{\theta}}) = \left\| F(\hat{\boldsymbol{\theta}}) - \mathbf{x} \right\|_2^2 \tag{2}$$

*then the parameter* $\boldsymbol{\theta}$ *is **fully-identified** (Defn. 3.1) by the estimator* $\hat{\boldsymbol{\theta}}$.

*Proof.* We begin by showing the global minimum of $\mathcal{L}(\hat{\boldsymbol{\theta}})$ exists and equals zero. Then, we show by contradiction that any estimators $\hat{\boldsymbol{\theta}}$ that obtains this global minimum has to equal the ground truth parameters $\boldsymbol{\theta}$.

**Step 1.** We show that the global minimum zero can be obtained for $\mathcal{L}(\hat{\boldsymbol{\theta}})$. Consider the ground truth parameter $\boldsymbol{\theta} \in \Theta$, then by definition of the ODE solver $F$ (Defn. 3.2), we have:

$$\mathcal{L}(\boldsymbol{\theta}) = \| F(\boldsymbol{\theta}) - \mathbf{x} \|_2^2 = \| \mathbf{x} - \mathbf{x} \|_2^2 = 0. \tag{6}$$

**Step 2.** Suppose for a contraction that there exists a $\boldsymbol{\theta}^* \in \Theta$ that minimizes the loss eq. (2) but differs from the ground truth parameters $\boldsymbol{\theta}$, i.e., $\boldsymbol{\theta}^* \neq \boldsymbol{\theta}$. This implies:

$$\mathcal{L}(\boldsymbol{\theta}^*) = \|F(\boldsymbol{\theta}^*) - \mathbf{x}\|_2^2 = 0 \tag{7}$$

Note that $\mathcal{L}(\boldsymbol{\theta}^*)$ can be rewritten as:

$$\mathcal{L}(\boldsymbol{\theta}^*) = \sum_{k=1}^{T} \|F(\boldsymbol{\theta}^*)_{t_k} - \mathbf{x}(t_k)\|_2^2 = 0 \tag{8}$$

To make sure the sum is zero, each individual term has to be zero, that is $F(\boldsymbol{\theta}^*)_{t_k} = \mathbf{x}(t_k), \forall t \in \{t_1, \ldots, t_T\}$. According to the uniqueness assumption of the ODE (Asm. 2.1), this implies $\boldsymbol{\theta}^* = \boldsymbol{\theta}$, which leads to a contradiction.

Thus, we have shown that minimizing eq. (2) will yield the ground truth parameter $\boldsymbol{\theta}$. In other words, any estimator $\hat{\boldsymbol{\theta}}$ that minimizes eq. (2) fully identifies $\boldsymbol{\theta}$. □

**Full identifiability with closed form solution when $f_{\boldsymbol{\theta}}$ is linear in $\boldsymbol{\theta}$.** We show that a closed-form solution can be obtained through linear least squares when the vector field $f_{\boldsymbol{\theta}}$ is linear in $\boldsymbol{\theta}$ and if we observe a *first-order* trajectory. A *first-order* trajectory means the first-order derivatives are included in the state-space vector. This statement is formalized as follows:

**Observation B.1.** Given a first-order trajectory $(\mathbf{x}, \dot{\mathbf{x}}) = (\mathbf{x}(t), \dot{\mathbf{x}}(t))_{t\in\mathcal{T}}$ generated from a dynamical system $f_{\boldsymbol{\theta}}(\mathbf{x}(t))$ satisfying Asms. 2.1 and 2.2. In particular, this ODE $f_{\boldsymbol{\theta}}$ can be written as a weighted sum of a set of base functions $\{\phi_1, \ldots, \phi_m\}$, i.e., $f_{\boldsymbol{\theta}}$ is linear in $\boldsymbol{\theta}$:

$$f_{\boldsymbol{\theta}}(\mathbf{x}(t)) = \sum_{i=1}^{m} \theta_i \phi_i(\mathbf{x}). \tag{9}$$

Define $\Phi_{\mathbf{x}} := [\phi_i(\mathbf{x}(t))]_{i\in[m], t\in\mathcal{T}} \in \mathbb{R}^{m\times T}$, then the global optimum of the loss eq. (2) is given by

$$\boldsymbol{\theta}^* = (\Phi_{\mathbf{x}}^{\intercal}\Phi_{\mathbf{x}})^{-1}\phi_{\mathbf{x}}\dot{\mathbf{x}} \tag{10}$$

As a direct implication, SINDy-like approaches [9, 10, 40] and gradient matching [68] can fully identify the underlying physical parameters $\boldsymbol{\theta}$ even with a closed-form solution if the underlying vector field $f_{\boldsymbol{\theta}}$ is can be represented as a sparse weighted sum of the given base functions $\{\phi_i\}_{i\in[m]}$.

## B.2 Proofs for partial identifiability

**Corollary 3.2** (Identifiability without known functional form). *Assume a dynamical system $f$ satisfying Asms. 2.1 and 2.2, a pair of trajectories $\mathbf{x}, \tilde{\mathbf{x}}$ generated from the same system $f$ but specified by different parameters $\boldsymbol{\theta}, \tilde{\boldsymbol{\theta}}$, respectively. Assume a partition of parameters $\boldsymbol{\theta}_S$ with $S \subseteq [N]$ is shared across the pair of parameters $\boldsymbol{\theta}, \tilde{\boldsymbol{\theta}}$. Let $g : \mathcal{X}^T \to \Theta$ be some smooth encoder and $\hat{F} : \Theta \to \mathcal{X}^T$ be some left-invertible smooth solver that minimizes the following objective:*

$$\mathcal{L}(g, \hat{F}) = \mathbb{E}_{\mathbf{x},\tilde{\mathbf{x}}} \underbrace{\|g(\mathbf{x})_S - g(\tilde{\mathbf{x}})_S\|_2^2}_{\textit{Alignment}} + \underbrace{\left\|\hat{F}(g(\mathbf{x})) - \mathbf{x}\right\|_2^2 + \left\|\hat{F}(g(\tilde{\mathbf{x}})) - \tilde{\mathbf{x}}\right\|_2^2}_{\textit{Sufficiency}}, \tag{3}$$

*then the shared partition $\boldsymbol{\theta}_S$ is partially identified (Defn. 3.3) by $g$ in the statistical setting.*

*Proof.* This proof can be directly adapted from the proofs with by Daunhawer et al. [16], Von Kügel-gen et al. [65], Yao et al. [71] with slight modification. So we briefly summarize the **Step 1.** and **Step 2.** that are imported from previous work and focus on the modification (**Step 3.**).

**Step 1.** We show that the loss function eq. (3) is lower bounded by zero and construct optimal encoder $g^* : \mathcal{X}^T \to \Theta$ that reach this lower bound. Define $g^* : \mathcal{X}^T \to \Theta := F^{-1}$ as the inverse of the ground truth data generating process, i.e., for all trajectories $\mathbf{x} = F(\boldsymbol{\theta})$ that generated from parameter $\boldsymbol{\theta}$, it holds:

$$g^*(\mathbf{x}) = \boldsymbol{\theta} \tag{11}$$

Thus, we have shown that the global minimum *zero* exists and can be obtained by the inverse mixing function $F^{-1} : \mathcal{X}^T \to \Theta$ (Defn. 3.2).

**Step 2.** We show that any optimal encoders $g$ that minimizes eq. (3) must have the *alignment* equal zero, in other words, it has to satisfy the ***invariance*** condition, which is formalized as

$$g(\mathbf{x})_S = g(\tilde{\mathbf{x}}) \qquad a.s. \tag{12}$$

Following Yao et al. [71, Lemma D.3], we conclude that both $g(\mathbf{x})_S$ and $g(\tilde{\mathbf{x}})_S$ can only depend on information about the shared partition about the ground truth parameter $\boldsymbol{\theta}_S$. In other words,

$$g(\mathbf{x})_S = g(\tilde{\mathbf{x}})_S = h(\boldsymbol{\theta}_S) \tag{13}$$

for some smooth $h : \Theta_S \to \Theta_S$.

**Step 3.** At last, we show that $h$ is invertible. Note that any optimal encoders $g$ that minimizes eq. (3) must have zero reconstruction error on both $\mathbf{x}$ and $\tilde{\mathbf{x}}$. Taking $\mathbf{x}$ as an example, we have

$$\mathbb{E}\left\|\hat{F}(g(\mathbf{x})) - \mathbf{x}\right\|_2^2 = 0 \tag{14}$$

which implies

$$\hat{F}(g(\mathbf{x})) = \mathbf{x} \qquad a.s. \tag{15}$$

If two continuous functions $\hat{F}(g(\mathbf{x}))$ and $\mathbf{x}$ equals *almost* everywhere on $\boldsymbol{\Theta}$, then they are equal everywhere on $\boldsymbol{\Theta}$, which implies:

$$\hat{F}(g(\mathbf{x})) = \mathbf{x} \qquad \forall \boldsymbol{\theta} \in \boldsymbol{\Theta} \tag{16}$$

Substituting $\mathbf{x}$ with the ground truth generating process $F$:

$$\hat{F}(g(\mathbf{x})) = F(\boldsymbol{\theta}) \qquad \forall \boldsymbol{\theta} \in \boldsymbol{\Theta}, \tag{17}$$

applying the left inverse of $\hat{F}$, we have:

$$\hat{F}^{-1} \circ \hat{F}(g(\mathbf{x})) = \hat{F}^{-1} \circ F(\boldsymbol{\theta}) \qquad \forall \boldsymbol{\theta} \in \boldsymbol{\Theta}, \tag{18}$$

i.e.,

$$g(\mathbf{x}) = \hat{F}^{-1} \circ F(\boldsymbol{\theta}) = \hat{F}^{-1} \circ F(\boldsymbol{\theta}_S, \boldsymbol{\theta}_{\bar{S}}) \qquad \forall \boldsymbol{\theta} \in \boldsymbol{\Theta}, \tag{19}$$

Define $h^* := \hat{F}^{-1} \circ F$, note that $h^*$ is bijective as a composition of bijections. Imposing the ***invariance*** constraint, we have $g(\mathbf{x})_S = h^*(\boldsymbol{\theta}_S, \boldsymbol{\theta}_{\bar{S}})_S$. Since $g(\mathbf{x})_S$ cannot depend on $\boldsymbol{\theta}_{\bar{S}}$ as shown in **Step 2**, we have $g(\mathbf{x})_S = h_S^*(\boldsymbol{\theta}_S)$ with $h := h_S^* : \boldsymbol{\Theta}_S \to \boldsymbol{\Theta}_S$.

Thus we have shown that $g(\mathbf{x})_S$ *partially* identifies $\boldsymbol{\theta}_S$.

$\square$

## C    Experimental results

**General remarks.** All models used in the experiments (§ 6) (*Ada-GVAE, TI-MNN, contrastive identifier, mechanistic identifier*) were built upon open-sourced code provided by the original works [39, 47, 71], under the MIT license. For *mechanistic identifiers*, we add a regularizer multiplier on the *alignment* constraint (Defn. 3.3), which is shown in Tabs. 3 and 5.

### C.1    Wind simulation: `SpeedyWeather.jl`

We simulate global air motion using using the `ShallowWaterModel` from `speedy weather` Julia package [28]. We consider a *layer thickness* as the primary generating factor in `ShallowWaterModel` varying from 8e3[m] to 2e4[m], which is a reasonable range given by the climate science literature. Taking the minimal and maximal values, we simulate the wind in a binary fashion and obtain 9024 trajectories across the globe under different conditions. Each trajectory constitutes three output variables discretized on `ts=121` time steps, on a 3D resolution grid of size: latitude `lat=47`; longitude `lon=96`; level `lev=1`. The three output variables represent *u wind component* (parallel to longitude), *v wind component* (parallel to latitude), and *relative vorticity*, respectively. An illustrative example of all three components is depicted in Fig. 5. further details about the simulation output are provided in Tab. 4. In particular, to train more efficiently, we pre-process the data using a *discrete cosine transform* (DCT) proposed by Ahmed et al. [1] and only keep the first 50% frequencies. This is feasible as the original data possesses a certain periodic pattern, as shown in Fig. 6.

For all baselines, we train the model till convergence. More training and test details for the tasks in § 6.2 are summarized in Tab. 3. To validate identifiability, we use `LogisticRegression` model from `scikit-learn` in its default setting to evaluate the classification accuracy in Fig. 3.

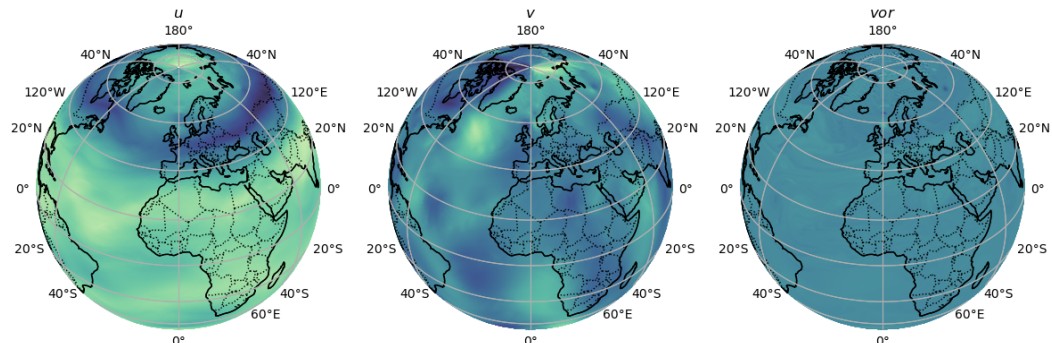

Figure 5: **Example of wind simulation**: *Left:* longitudinal wind velocity ($u$) [m/s]. *Middle*: latitudinal wind velocity ($v$)[m/s], *Right*: relative vorticity ($vor$) [1/s].

Table 3: Training setup for wind simulation in § 6.2. Non-applicable fields are marked with ✗.

|  | Ada-GVAE | TI-MNN | Cont. Identifier | Mech. Identifier |
|---|---|---|---|---|
| Pre-process | DCT | DCT | DCT | DCT |
| Encoder | 6-layer MLP | 6-layer MLP | 6-layer MLP | 6-layer MLP |
| Decoder | 6-layer MLP | 6-layer MLP | ✗ | 3 proj. $\times$ 6-layer MLP |
| Time dim | 121 | 121 | 121 | 121 |
| State dim | 2 | 2 | 2 | 2 |
| Hidden dim | 1024 | 1024 | 1024 | 1024 |
| Latent dim | 12 | 12 | 12 | 12 |
| Optimizer | Adam | Adam | Adam | Adam |
| Adam: learning rate | 1e−5 | 1e−5 | 1e−5 | 1e−5 |
| Adam: beta1 | 0.9 | 0.9 | 0.9 | 0.9 |
| Adam: beta2 | 0.999 | 0.999 | 0.999 | 0.999 |
| Adam: epsilon | 1e−8 | 1e−8 | 1e−8 | 1e−8 |
| Batch size | 1128 | 1128 | 1128 | 1128 |
| Temperature $\tau$ | ✗ | ✗ | 0.1 | ✗ |
| Alignment reg. | ✗ | ✗ | ✗ | 10 |
| # Initial values | 10 | 10 | ✗ | 10 |
| # Iterations | < 30,000 | < 30,000 | < 30,000 | < 30,000 |
| # Seeds | 3 | 3 | 3 | 3 |

## C.2 Sea surface temperature: SST-V2

The sea surface temperature data SST-V2 [21] contains the *weekly* sea surface temperature data from 1990 to 2023, on a resolution grid of $180 \times 360$ (latitudes $\times$ longitudes). An example input is depicted in Fig. 8. Each time series contains 1727 times steps. To generate multiple views that share specific climate properties, we sample two different trajectories from a small neighbor region ($\pm 5°$) along the ***same latitude***, as the latitude differs in the amount of direct sunlight thus directly affecting the sea surface temperature.

For a fair comparison, we train all baselines till convergence following the setup summarized in Tab. 5. Similar to the wind simulation data, we pre-process the SST-V2 data using DCT and keep the first 25% frequencies, as the latitude-related parameters of interest primarily influence long-term dependencies, such as seasonality, which are predominantly captured by low-frequency components. Fig. 7 shows an example of predicted trajectories over three randomly sampled locations. As for the downstream classification task, we use `LogisticRegression` model from `scikit-learn` in its default setting to evaluate the classification accuracy in Tab. 2.

## C.3 Experiments and computational resources

In this paper, we train four different models, each over three independent seeds. All 12 jobs ran with 24GB of RAM, 8 CPU cores, and a single node GPU, which is, in most cases, `NVIDIA GeForce RTX2080Ti`. Given different model sizes and convergence rates, the required amount of compute

Table 4: **Wind simulation**: output variables.

| Output variable [unit] | Shape |
|---|---|
| Longitudinal wind velocity ($u$) [m/s] | (`ts`, `lev`, `lat`, `lon`) |
| Latitudinal wind velocity ($v$) [m/s] | (`ts`, `lev`, `lat`, `lon`) |
| Relative vorticity ($vor$) [1/s] | (`ts`, `lev`, `lat`, `lon`) |

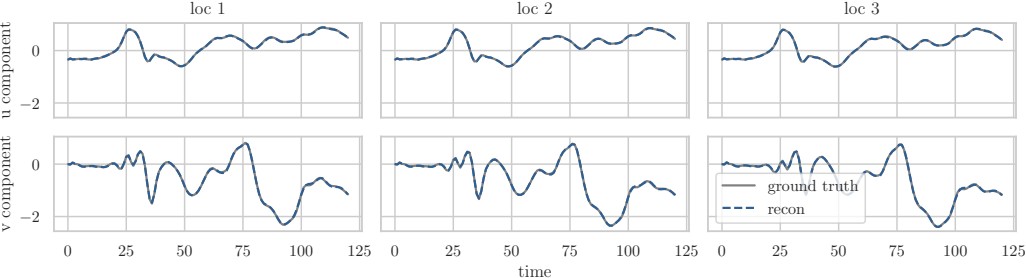

Figure 6: **Wind simulation**: *mechanistic identifier* reconstruction of highly irregular time series. The first half of the trajectory is provided as initial values, while the second half is predicted.

could vary slightly, despite the pre-fixed training epochs. Thus, we report an upper bound of the compute hours on `NVIDIA GeForce RTX2080Ti`. On average, all runs converge within 22 GPU hours. Therefore, the experimental results in this paper can be reproduced with 264 GPU hours.

## D Discussion

**Why mechanistic neural networks [47]**. As mentioned in § 4, the ODE solver $F$ given in Cors. 3.1 and 3.2 can be interpreted as the decoder in a traditional representation learning regime; however, several challenges arise when integrating ODE solving in the training loop: First of all, the ODE solver must be differentiable to utilize the automatic differentiation implementation of the state-of-the-art deep learning frameworks; this obstacle has been tacked by the line of work termed *NeuralODE*, which models the ODE vector field using a neural network thus enable differentiability [12, 13, 25]. Nevertheless, most differentiable ODE solvers solve the ODE autoregressively and thus cannot be parallelized by the GPU very efficiently. Dealing with long-term trajectories (for example, weekly climate data during the last few decades) would be extremely computationally heavy. Therefore, we advocate for a time- and memory-efficient differentiable ODE solver: the mechanistic neural networks [47].

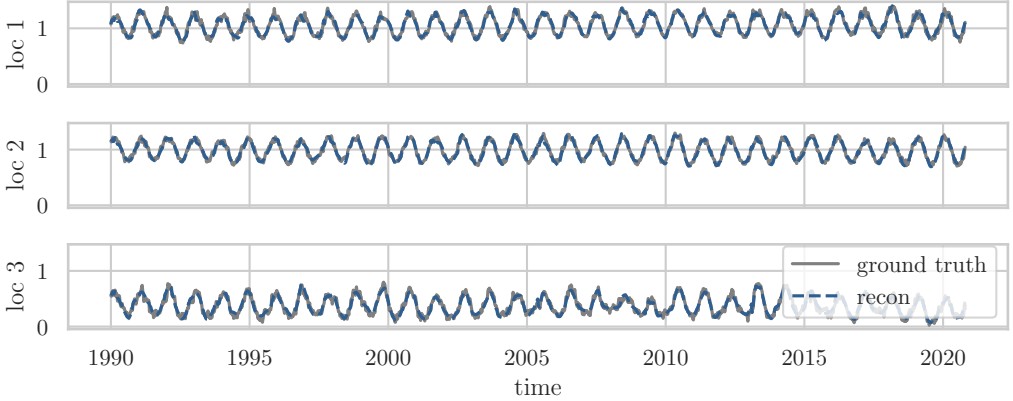

Figure 7: **SST-V2**: *mechanistic identifier* reconstruction over long-term time series. Results are produced by concatenating subsequently predicted chunks.

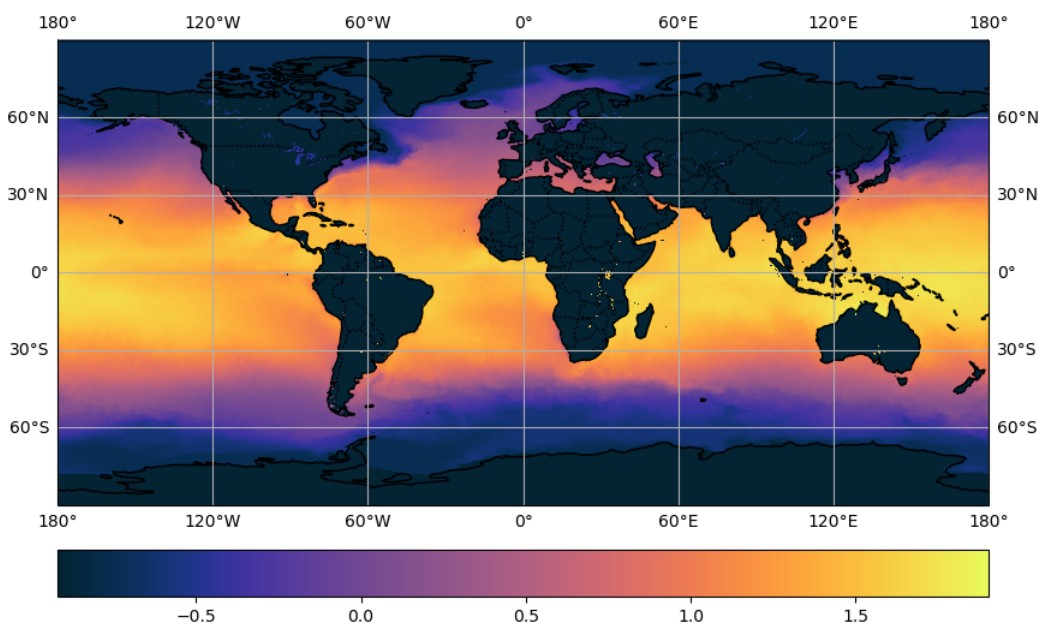

Figure 8: **Example of global sea surface temperature** in January, 1990.

Table 5: Training setup for sea surface temperature in § 6.3. Non-applicable fields are marked with ✗.

|  | Ada-GVAE | TI-MNN | Cont. Identifier | Mech. Identifier |
|---|---|---|---|---|
| Pre-process | DCT | DCT | DCT | DCT |
| Encoder | 6-layer MLP | 6-layer MLP | 6-layer MLP | 6-layer MLP |
| Decoder | 6-layer MLP | 6-layer MLP | ✗ | 3 proj. $\times$ 6-layer MLP |
| Time dim | 208 | 208 | 208 | 208 |
| State dim | 1 | 1 | 1 | 1 |
| Hidden dim | 1024 | 1024 | 1024 | 1024 |
| Latent dim | 20 | 20 | 20 | 20 |
| Optimizer | Adam | Adam | Adam | Adam |
| Adam: learning rate | 1e−5 | 1e−5 | 1e−5 | 1e−5 |
| Adam: beta1 | 0.9 | 0.9 | 0.9 | 0.9 |
| Adam: beta2 | 0.999 | 0.999 | 0.999 | 0.999 |
| Adam: epsilon | 1e−8 | 1e−8 | 1e−8 | 1e−8 |
| Batch size | 2160 | 2160 | 2160 | 2160 |
| Temperature $\tau$ | ✗ | ✗ | 0.1 | ✗ |
| Alignment reg. | ✗ | ✗ | ✗ | 10 |
| # Initial values | 10 | 10 | ✗ | 10 |
| # Iterations | < 30,000 | < 30,000 | < 30,000 | < 30,000 |
| # Seeds | 3 | 3 | 3 | 3 |

**Latent regularizers in CRL.** The framework proposed in § 4 can be generalized to many causal representation learning works, by specifying the latent regularizes according to individual assumptions and settings. For example, in the multiview setting, the latent regularizer can be the $L_2$ *alignment* between the learned representations on the shared partition eq. (3), as it was assumed that the paired views are generated based on this overlapping set of latents [38, 65, 71]; in sparse causal representation learning the underlying generative process assumes observations are generated from sparse latent variables; therefore, the proposed algorithms actively enforce some sparsity constraint on the learned representation [29, 31, 45, 70], We provide a more extensive summary of different causal representation learning approaches and their corresponding latent regularizer in Tab. 6. By replacing the *alignment* term (Cor. 3.2) with the specific latent constraints, one can plug in many causal representation learning algorithms to construct an identifiable neural emulator using our framework.

Table 6: A non-exhaustive summary of latent regularizers in recent CRL approaches.

| Principle | Assumption | Latent regularizer | References |
|---|---|---|---|
| *multiview* | *part. shared* latents | $\|g(\mathbf{x})_S - g(\tilde{\mathbf{x}})_S\|_2^2$ | Locatello et al. [39], Von Kügelgen et al. [65] |
| | | | Daunhawer et al. [16], Yao et al. [71] |
| | | $\|g(\tilde{\mathbf{x}}) - g(\mathbf{x}) - \delta\|_2^2$ | Ahuja et al. [2] |
| *sparsity* | *sparse* causal graph | $\|g(\mathbf{x})\|_1$ | Lachapelle et al. [31], Xu et al. [70] |
| | | Spike and Slab prior | Moran et al. [45], Tonolini et al. [61] |
| | temporal sparsity | $\text{KL}\left(q(z^t \mid x^t)\|\hat{p}(z^t\|z^{<t}, a^{<t})\right)$ | Lachapelle and Lacoste-Julien [29] |

**Identifying time-varying parameters** Time-varying parameters $\boldsymbol{\theta}(t)$ could also be potentially identified when they change sparsely in time. For example, a time-varying parameter $\boldsymbol{\theta}_k$ remains constant between $(t_k, t_{k+1})$. Then, the states in between $\mathbf{x}(t), \mathbf{x}(t+1), \ldots, \mathbf{x}(t+k)$ can be considered as multiple views that share the same parameter $\boldsymbol{\theta}_k$. Following this perspective, the time-invariant parameters considered in the scope of this paper remain consistent through the whole timespan $(0, t_{\max}$, thus all discretized states $\mathbf{x}(t_1), \ldots, \mathbf{x}(t_T)$ are views that share this parameter. This inductive bias is directly built into the architecture design by inputting the whole trajectory into the encoder instead of doing so step by step (where the time axis is considered as batch dimension). From another angle, the time-varying parameters $\boldsymbol{\theta}(t)$ could be interpreted as a *hidden* part of the state space vector $\mathbf{x}(t)$ without an explicitly defined differential equation, which gives rise to a partial observable setup. This direction has been studied in the context of sparse system identification without explicit identifiability analysis [40]. In a more general setting, time-varying parameters has been considered as latent trajectories and extensively studied in the field of temporal causal representation learning [33, 57, 72]. A more detailed related work section in this regard is provided in § 5 under **CRL and dynamical systems**.

**Model evaluation on real-world data.** A great obstacle hindering causal representation learning scaling to real-world data is that no ground truth latent variables are available. Since the methods aim to *identify* the latent variables, it is hard to validate the identifiability theory without ground truth-generating factors. However, properly evaluating the CRL models on real-world data can be conducted by carefully designing causal downstream tasks, such as climate zone classification and ATE estimation shown in § 6.3. Overall, we believe by incorporating domain knowledge of the applied datasets, we can use CRL to answer important causal questions from individual fields, thus indirectly validating the identifiability.

Table 7: **Theorem 3.1 validation using ODEs with known functional form**: Experiments on complex dynamical systems from ODEBench [15] and Cart-Pole (inspired by Yao et al. [72]), for **exact parameter identification**. RMSE is computed over 100 randomly sampled parameter groups (nearby chaotic configuration for chaotic systems) and averaged over the parameter dimension.

| ID | System description | Equation | RMSE (m ± std) |
|---|---|---|---|
| 1 | RC-circuit (charging capacitor) | $\frac{\theta_0 - \frac{x_0}{\theta_1}}{\theta_2}$ | $3e-2 \pm 2e-2$ |
| 2 | Population growth (naive) | $\theta_0 x_0$ | $2e-5 \pm 9e-6$ |
| 3 | Population growth with carrying capacity | $\theta_0 x_0 \cdot \left(1 - \frac{x_0}{\theta_1}\right)$ | $4e-5 \pm 2e-6$ |
| 4 | RC-circuit with non-linear resistor (charging capacitor) | $-0.5 + \frac{1}{e^{\theta_0 - \frac{x_0}{\theta_1}} + 1}$ | $8e-5 \pm 4e-4$ |
| 5 | Velocity of a falling object with air resistance | $\theta_0 - \theta_1 x_0^2$ | $6e-4 \pm 4e-4$ |
| 6 | Autocatalysis with one fixed abundant chemical | $\theta_0 x_0 - \theta_1 x_0^2$ | $9e-5 \pm 1e-5$ |
| 7 | Gompertz law for tumor growth | $\theta_0 x_0 \log(\theta_1 x_0)$ | $3e-2 \pm 4e-2$ |
| 8 | Logistic equation with Allee effect | $\theta_0 x_0 \left(-1 + \frac{x_0}{\theta_2}\right)\left(1 - \frac{x_0}{\theta_1}\right)$ | $8e-3 \pm 9e-3$ |
| 9 | Language death model for two languages | $\theta_0 \cdot (1 - x_0) - \theta_1 x_0$ | $1e-4 \pm 3e-5$ |
| 10 | Refined language death model for two languages | $\theta_0 x_0^{\theta_1} \cdot (1 - x_0) - x_0 \cdot (1 - \theta_0)(1 - x_0)^{\theta_1}$ | $1e-5 \pm 2e-5$ |
| 11 | Naive critical slowing down (statistical mechanics) | $-x_0^3$ | ✗ |
| 12 | Photons in a laser (simple) | $\theta_0 x_0 - \theta_1 x_0^2$ | $4e-3 \pm 3e-3$ |
| 13 | Overdamped bead on a rotating hoop | $\theta_0 (\theta_1 \cos(x_0) - 1) \sin(x_0)$ | $3e-4 \pm 7e-5$ |
| 14 | Budworm outbreak model with predation | $\theta_0 x_0 \cdot \left(1 - \frac{x_0}{\theta_1}\right) - \frac{\theta_3 x_0^2}{\theta_2^2 + x_0^2}$ | $5e-3 \pm 9e-4$ |
| 15 | Budworm outbreak with predation (dimensionless) | $\theta_0 x_0 \cdot \left(1 - \frac{x_0}{\theta_1}\right) - \frac{x_0^2}{x_0^2 + 1}$ | $4e-5 \pm 5e-6$ |
| 16 | Landau equation (typical time scale tau = 1) | $\theta_0 x_0 - \theta_1 x_0^3 - \theta_2 x_0^5$ | $5e-3 \pm 1e-2$ |
| 17 | Logistic equation with harvesting/fishing | $\theta_0 x_0 \cdot \left(1 - \frac{x_0}{\theta_1}\right) - \theta_2$ | $6e-4 \pm 2e-4$ |
| 18 | Improved logistic equation with harvesting/fishing | $\theta_0 x_0 \cdot \left(1 - \frac{x_0}{\theta_1}\right) - \frac{\theta_2 x_0}{\theta_3 + x_0}$ | $4e-2 \pm 2e-2$ |
| 19 | Improved logistic equation with harvesting/fishing (dimensionless) | $-\frac{\theta_0 x_0}{\theta_1 + x_0} + x_0 \cdot (1 - x_0)$ | $4e-5 \pm 2e-5$ |
| 20 | Autocatalytic gene switching (dimensionless) | $\theta_0 - \theta_1 x_0 + \frac{x_0^2}{x_0^2 + 1}$ | $2e-5 \pm 1e-5$ |
| 21 | Dimensionally reduced SIR infection model for dead people (dimensionless) | $\theta_0 - \theta_1 x_0 - e^{-x_0}$ | $8e-6 \pm 2e-6$ |
| 22 | Hysteretic activation of a protein expression (positive feedback, basal promoter expression) | $\theta_0 + \frac{\theta_1 x_0^5}{\theta_2 + x_0^5} - \theta_3 x_0$ | $3e-2 \pm 2e-2$ |
| 23 | Overdamped pendulum with constant driving torque/fireflies/Josephson junction (dimensionless) | $\theta_0 - \sin(x_0)$ | $8e-6 \pm 8e-7$ |
| 24 | Harmonic oscillator without damping | $\begin{cases} x_1 \\ -\theta_0 x_0 \end{cases}$ | $4e-4 \pm 2e-5$ |
| 25 | Harmonic oscillator with damping | $\begin{cases} x_1 \\ -\theta_0 x_0 - \theta_1 x_1 \end{cases}$ | $9e-4 \pm 1e-4$ |
| 26 | Lotka-Volterra competition model (Strogatz version with sheeps and rabbits) | $\begin{cases} x_0(\theta_0 - \theta_1 x_1 - x_0) \\ x_1(\theta_2 - x_0 - x_1) \end{cases}$ | $7e-2 \pm 4e-2$ |
| 27 | Lotka-Volterra simple (as on Wikipedia) | $\begin{cases} x_0(\theta_0 - \theta_1 x_1) \\ -x_1(\theta_2 - \theta_3 x_0) \end{cases}$ | $9e-3 \pm 1e-3$ |
| 28 | Pendulum without friction | $\begin{cases} x_1 \\ -\theta_0 \sin(x_0) \end{cases}$ | $6e-5 \pm 2e-5$ |
| 29 | Dipole fixed point | $\begin{cases} \theta_0 x_0 x_1 \\ -x_0^2 + x_1^2 \end{cases}$ | $2e-3 \pm 3e-4$ |

| ID | System description | Equation | RMSE (m ± std) |
|---|---|---|---|
| 30 | RNA molecules catalyzing each others replication | $\begin{cases} x_0(-\theta_0 x_0 x_1 + x_1) \\ x_1(-\theta_0 x_0 x_1 + x_0) \end{cases}$ | $7e{-}6 \pm 4e{-}7$ |
| 31 | SIR infection model only for healthy and sick | $\begin{cases} -\theta_0 x_0 x_1 \\ \theta_0 x_0 x_1 - \theta_1 x_1 \end{cases}$ | $4e{-}5 \pm 2e{-}5$ |
| 32 | Damped double well oscillator | $\begin{cases} x_1 \\ -\theta_0 x_1 - x_0^3 + x_0 \end{cases}$ | $3e{-}4 \pm 5e{-}5$ |
| 33 | Glider (dimensionless) | $\begin{cases} -\theta_0 x_0^2 - \sin(x_1) \\ x_0 - \frac{\cos(x_1)}{x_0} \end{cases}$ | $3e{-}4 \pm 2e{-}4$ |
| 34 | Frictionless bead on a rotating hoop (dimensionless) | $\begin{cases} x_1 \\ (-\theta_0 + \cos(x_0))\sin(x_0) \end{cases}$ | $4e{-}5 \pm 2e{-}5$ |
| 35 | Rotational dynamics of an object in a shear flow | $\begin{cases} \cos(x_0)\cot(x_1) \\ \left(\theta_0 \sin^2(x_1) + \cos^2(x_1)\right)\sin(x_0) \end{cases}$ | $3e{-}3 \pm 3e{-}4$ |
| 36 | Pendulum with non-linear damping, no driving (dimensionless) | $\begin{cases} x_1 \\ -\theta_0 x_1 \cos(x_0) - x_1 - \sin(x_0) \end{cases}$ | $5e{-}4 \pm 1e{-}4$ |
| 37 | Van der Pol oscillator (standard form) | $\begin{cases} x_1 \\ -\theta_0 x_1 \left(x_0^2 - 1\right) - x_0 \end{cases}$ | $4e{-}4 \pm 6e{-}5$ |
| 38 | Van der Pol oscillator (simplified form from Strogatz) | $\begin{cases} \theta_0\left(-\frac{x_0^3}{3} + x_0 + x_1\right) \\ -\frac{x_0}{\theta_0} \end{cases}$ | $2e{-}3 \pm 1e{-}4$ |
| 39 | Glycolytic oscillator, e.g., ADP and F6P in yeast (dimensionless) | $\begin{cases} \theta_0 x_1 + x_0^2 x_1 - x_0 \\ -\theta_0 x_0 + \theta_1 - x_0^2 x_1 \end{cases}$ | $8e{-}4 \pm 7e{-}5$ |
| 40 | Duffing equation (weakly non-linear oscillation) | $\begin{cases} x_1 \\ \theta_0 x_1 \cdot \left(1 - x_0^2\right) - x_0 \end{cases}$ | $9e{-}4 \pm 1e{-}4$ |
| 41 | Cell cycle model by Tyson for interaction between protein cdc2 and cyclin (dimensionless) | $\begin{cases} \theta_0\left(\theta_1 + x_0^2\right)(-x_0 + x_1) - x_0 \\ \theta_2 - x_0 \end{cases}$ | $4e{-}2 \pm 2e{-}2$ |
| 42 | Reduced model for chlorine dioxide-iodine-malonic acid reaction (dimensionless) | $\begin{cases} \theta_0 - \frac{\theta_1 x_0 x_1}{x_0^2 + 1} - x_0 \\ \theta_2 x_0(-\frac{x_1}{x_0^2 + 1} + 1) \end{cases}$ | $3e{-}3 \pm 3e{-}4$ |
| 43 | Driven pendulum with linear damping / Josephson junction (dimensionless) | $\begin{cases} x_1 \\ \theta_0 - \theta_1 x_1 - \sin(x_0) \end{cases}$ | $6e{-}5 \pm 3e{-}5$ |
| 44 | Driven pendulum with quadratic damping (dimensionless) | $\begin{cases} x_1 \\ \theta_0 - \theta_1 x_1 |x_1| - \sin(x_0) \end{cases}$ | $2e{-}5 \pm 6e{-}6$ |
| 45 | Isothermal autocatalytic reaction model by Gray and Scott 1985 (dimensionless) | $\begin{cases} \theta_0 \cdot (1 - x_0) - x_0 x_1^2 \\ -\theta_1 x_1 + x_0 x_1^2 \end{cases}$ | $2e{-}5 \pm 3e{-}5$ |
| 46 | Interacting bar magnets | $\begin{cases} \theta_0 \sin(x_0 - x_1) - \sin(x_0) \\ -\theta_0 \sin(x_0 - x_1) - \sin(x_1) \end{cases}$ | $1e{-}6 \pm 1e{-}6$ |
| 47 | Binocular rivalry model (no oscillations) | $\begin{cases} -x_0 + \frac{1}{e^{\theta_0 x_1 - \theta_1} + 1} \\ -x_1 + \frac{1}{e^{\theta_0 x_0 - \theta_1} + 1} \end{cases}$ | $7e{-}4 \pm 1e{-}4$ |
| 48 | Bacterial respiration model for nutrients and oxygen levels | $\begin{cases} \theta_0 - \frac{x_0 x_1}{\theta_1 x_0^2 + 1} - x_0 \\ \theta_2 - \frac{x_0 x_1}{\theta_1 x_0^2 + 1} \end{cases}$ | $4e{-}2 \pm 2e{-}2$ |
| 49 | Brusselator: hypothetical chemical oscillation model (dimensionless) | $\begin{cases} \theta_1 x_0^2 x_1 - x_0(\theta_0 + 1) + 1 \\ \theta_0 x_0 - \theta_1 x_0^2 x_1 \end{cases}$ | $2e{-}2 \pm 5e{-}3$ |
| 50 | Chemical oscillator model by Schnackenberg 1979 (dimensionless) | $\begin{cases} \theta_0 + x_0^2 x_1 - x_0 \\ \theta_1 - x_0^2 x_1 \end{cases}$ | $1e{-}6 \pm 4e{-}7$ |
| 51 | Oscillator death model by Ermentrout and Kopell 1990 | $\begin{cases} \theta_0 + \sin(x_1)\cos(x_0) \\ \theta_1 + \sin(x_1)\cos(x_0) \end{cases}$ | $1e{-}5 \pm 9e{-}6$ |

| ID | System description | Equation | RMSE (m $\pm$ std) |
|---|---|---|---|
| 52 | Maxwell-Bloch equations (laser dynamics) | $\begin{cases} \theta_0(-x_0+x_1) \\ \theta_1(x_0x_2-x_1) \\ \theta_2(-\theta_3x_0x_1+\theta_3-x_2+1) \end{cases}$ | $4e{-}2\pm4e{-}2$ |
| 53 | Model for apoptosis (cell death) | $\begin{cases} \theta_0-\theta_4x_0-\frac{\theta_5x_0x_1}{\theta_9+x_0} \\ \theta_1x_2(\theta_8+x_1)-\frac{\theta_2x_1}{\theta_6+x_1}-\frac{\theta_3x_0x_1}{\theta_7+x_1} \\ -\theta_1x_2(\theta_8+x_1)+\frac{\theta_2x_1}{\theta_6+x_1}+\frac{\theta_3x_0x_1}{\theta_7+x_1} \end{cases}$ | $1e{-}2\pm5e{-}3$ |
| 54 | Lorenz equations in well-behaved periodic regime | $\begin{cases} \theta_0(-x_0+x_1) \\ \theta_1x_0-x_0x_2-x_1 \\ -\theta_2x_2+x_0x_1 \end{cases}$ | $1e{-}2\pm4e{-}3$ |
| 55 | Lorenz equations in complex periodic regime | $\begin{cases} \theta_0(-x_0+x_1) \\ \theta_1x_0-x_0x_2-x_1 \\ -\theta_2x_2+x_0x_1 \end{cases}$ | $8e{-}2\pm5e{-}2$ |
| 56 | Lorenz equations (chaotic) | $\begin{cases} \theta_0(-x_0+x_1) \\ \theta_1x_0-x_0x_2-x_1 \\ -\theta_2x_2+x_0x_1 \end{cases}$ | $4e{-}2\pm9e{-}3$ |
| 57 | Rössler attractor (stable fixed point) | $\begin{cases} \theta_3(-x_1-x_2) \\ \theta_3(\theta_0x_1+x_0) \\ \theta_3(\theta_1+x_2(-\theta_2+x_0)) \end{cases}$ | $3e{-}2\pm2e{-}2$ |
| 58 | Rössler attractor (periodic) | $\begin{cases} \theta_3(-x_1-x_2) \\ \theta_3(\theta_0x_1+x_0) \\ \theta_3(\theta_1+x_2(-\theta_2+x_0)) \end{cases}$ | $2e{-}2\pm2e{-}2$ |
| 59 | Rössler attractor (chaotic) | $\begin{cases} \theta_3(-x_1-x_2) \\ \theta_3(\theta_0x_1+x_0) \\ \theta_3(\theta_1+x_2(-\theta_2+x_0)) \end{cases}$ | $4e{-}2\pm4e{-}2$ |
| 60 | Aizawa attractor (chaotic) | $\begin{cases} -\theta_3x_1+x_0(-\theta_1+x_2) \\ \theta_3x_0+x_1(-\theta_1+x_2) \\ \theta_0x_2+\theta_2+\theta_5x_0^3x_2-1/3x_2^3-\left(x_0^2+x_1^2\right)(\theta_4x_2+1) \end{cases}$ | $8e{-}4\pm2e{-}4$ |
| 61 | Chen-Lee attractor (chaotic) | $\begin{cases} \theta_0x_0-x_1x_2 \\ \theta_1x_1+x_0x_2 \\ \theta_2x_2+\frac{x_0x_1}{\theta_3} \end{cases}$ | $4e{-}2\pm2e{-}2$ |
| 62 | Binocular rivalry model with adaptation (oscillations) | $\begin{cases} -x_0+\frac{1}{e^{\theta_0x_2+\theta_1x_1-\theta_2}+1} \\ \theta_3(x_0-x_1) \\ -x_2+\frac{1}{e^{\theta_0x_0+\theta_1x_3-\theta_2}+1} \\ \theta_3(x_2-x_3) \end{cases}$ | $1e{-}3\pm2e{-}4$ |
| 63 | SEIR infection model (proportions) | $\begin{cases} -\theta_1x_0x_2 \\ -\theta_0x_1+\theta_1x_0x_2 \\ \theta_0x_1-\theta_2x_2 \\ \theta_2x_2 \end{cases}$ | $1e{-}4\pm5e{-}5$ |
| - | Cart-Pole (inverted pendulum) | $\begin{cases} \ddot{x} = \frac{F+m_pl(\dot{\alpha}^2\sin(\alpha)-\ddot{\alpha}\cos(\alpha))}{m_c+m_p} \\ \ddot{\alpha} = \frac{g\sin(\alpha)-\cos(\alpha)\frac{F+m_pl\dot{\alpha}^2\sin(\alpha)}{m_c+m_p}}{l\left(\frac{4}{3}-\frac{m_p\cos^2(\alpha)}{m_c+m_p}\right)} \end{cases}$ | $1e{-}4\pm6e{-}05$ |

