# OpenReview forum: "Marrying Causal Representation Learning with Dynamical Systems for Science"
_NeurIPS.cc/2024/Conference — NeurIPS 2024 poster_

### Official Review · Reviewer_yeqY · 2024-07-10

**Soundness:** 2
**Presentation:** 2
**Contribution:** 2
**Rating:** 5
**Confidence:** 3

**Summary:**

This paper aims to connect causal representation learning with parameter identification in dynamical systems. By doing so, existing causal representation learning approaches can be used for estimating parameter in dynamical systems with identification guarantees. On the other side, it also demonstrates the applicability of causal representation learning on real-world data. Experimental evaluation on simulated and real-world climate data show the effectiveness of applying causal representation learning approaches in solving parameter identification tasks in dynamical systems.

**Strengths:**

This paper focuses on an intriguing and emerging topic on parameter identification in dynamic systems. Particularly, the authors attempt to leverage existing causal representation approaches to address the parameter identification problem. Re-formulating parameter identification problem as a causal representation learning problem is an appealing insight.

**Weaknesses:**

1: Contribution is not significant. Though the paper aims to set up the connection between CRL and parameter estimation, most of the discussions are based on CRL. This work is more like an application of CRL to parameter estimation in dynamical systems. The identifiability theories are mostly adopted from CRL. It is hard to understand their practical effectiveness in solving parameter identification task in dynamic systems. Providing specific dynamic systems as examples can be helpful in understanding the usage of these theories.

2: Missing important related works. Parameter identification has been addressed with identifiability guarantee, e.g., [a], while these works are not well discussed.

[a] Yao, W., Chen, G., & Zhang, K. (2022). Temporally disentangled representation learning. Advances in Neural Information Processing Systems, 35, 26492-26503.

3: Experiments are insufficient. Only simulated wind data and real-data on sea surface temperature are considered. To have a comprehensive understanding of the effectiveness in dynamic systems, different types of dynamic systems should be considered. Ablation studies should be performed on scenarios where true functional form is known for verification.

**Questions:**

1: Are the assumptions (e.g., assumption 2.1 and 2.2) wild? Given which properties the dynamic systems can satisfy these assumptions?

2: How is your work different from [a], which can deal with videos and noisy data under identification guarantee?

3: Why you use mechanistic neural networks?

4: How does the proposed approach handle heterogeneous data with multiple sets of parameters?

**Limitations:**

The authors have discussed the limitations of the work.

---

> ### Author Rebuttal · Authors · 2024-08-05
>
> ### Reply to weaknesses (same order as given by the reviewer)
>
> 1. The [global rebuttal](https://openreview.net/forum?id=MWHRxKz4mq&noteId=n59QJ5gxJF) clarifies our paper’s **novelty and contribution**. Additionally, we successfully demonstrated parameter identification in various existing systems. Please refer to `additional experiments` in the [global rebuttal](https://openreview.net/forum?id=MWHRxKz4mq&noteId=n59QJ5gxJF).
> 2. We thank the reviewer for providing this line of related work ([a] is listed as [`8`] under our references). Our paper looks **superficially similar** to those, but they are **conceptually different**. Please allow us a quick clarification in this regard:
>    1. Reference [`8`] models dynamics in the **latent space**, whereas our approach models dynamics directly in the **observational** space. The reasons for this choice are detailed in the [global rebuttal](https://openreview.net/forum?id=MWHRxKz4mq&noteId=n59QJ5gxJF).
>    2. Consequently, our framework addresses **a different hierarchy of problems**. For instance, in the cart-pole example from Figure 3 in [`8`], while [`8`] treats the time-varying ODE states (**cart position, pole angle**) as latent variables, our focus is on estimating higher-level, time-invariant parameters such as **gravity, cart mass, pole mass, and pole length** for the known system (see `additional experiments`). A similar distinction applies to the motion capture dataset in [`8`].
>    3. Hence, we believe that both problems are **orthogonal yet equally important, and we encourage cross-pollination in future work**. We will thoroughly discuss these works in the updated manuscript and explore the potential applications of these approaches to scientific discoveries.
> 3. **Additional studies** are provided on systems with **known functional forms**, including the **cart-pole** system inspired by [`8`]. Please refer to `additional experiments` for details. We thank the reviewer for this suggestion and kindly ask them to reconsider their scores in light of the new experiments. While we cannot directly compare against [`8`] because we take as input the whole trajectory instead of modeling the dynamics step by step, we will carefully explain the differences and opportunities for synergy between the approaches.
>
> &nbsp;
> ### Reply to questions (same order as given by the reviewer)
>
> 1. The **validity of the assumptions** is discussed in the [global rebuttal](https://openreview.net/forum?id=MWHRxKz4mq&noteId=n59QJ5gxJF).
> 2. The **difference** between our paper and the **provided reference** is discussed in the second point of “Reply to weaknesses.”
> 3. The reason **why we use MNN** is discussed in Appendix D (`lines 727-739`). We will refer to this link more clearly in the updated manuscript, and we apologize that this was not clearly referenced in the main text of the submitted draft.
> 4. Thank you for this intriguing question. It is indeed **possible** to identify latent parameters from **heterogeneous** time series data using multiview CRL algorithms [`6`], which assume multimodal mixing functions for different views. This paper primarily focuses on a single climate dataset to **establish a connection** between causal representation learning and dynamical systems; thus, the consideration of heterogeneous data is **outside** **the** **scope** of this study.  However, we acknowledge that extending this approach to **heterogeneous** data is a **promising** area for future research. Broadly, we believe that many algorithms developed in causal representation learning can be adapted for dynamical systems, and we hope our paper lays the **foundations** for and **encourages** this future work.

---

> > ### Author Response · Authors · 2024-08-12
> > **We kindly ask if the rebuttal has addressed the reviewer's concerns**
> >
> > Dear reviewer `yeqY`:
> >
> > We greatly appreciate the reviewer's constructive feedback and thoughtful consideration. As the discussion period concludes soon, we kindly ask if the rebuttal has addressed their concerns.
> >
> > Should any further questions or concerns arise, we are happy to provide clarification during the remaining discussion time.

---

> > > ### Comment · Reviewer_yeqY · 2024-08-12
> > >
> > > Thanks for the authors' responses. My concerns have been addressed. I would like to increase my score.

---

> ### Comment · Area_Chair_e8B7 · 2024-08-12
> **Author-Reviewer Discussion Reminder**
>
> Dear reviewers
>
> Since we are approaching the end of review-author discussion. Please read the author's response and clearly acknowledge that if they have successfully addressed your concerns.
>
> Best,
>
> You AC

---

### Official Review · Reviewer_aFgN · 2024-07-12

**Soundness:** 3
**Presentation:** 3
**Contribution:** 3
**Rating:** 6
**Confidence:** 3

**Summary:**

The authors draw a connection between assumptions in the (neural) ODE and causal representation learning (more precisely, latent variable identifiability, but I will follow the authors and refer to it as CRL) literatures, by framing ODE inverse problems as latent variable problems. A particular focus is given to the case where the parametric form of the ODE is unknown, and only trajectories (i.e., time series data) are given. The learned generator and latent variables in CRL are analogous to a flexibly parametrized ODE solver and corresponding parameters, respectively. An example is given where pairs of trajectories sharing some parameter values are interpreted as same-content-different-style observations, which can be block-identified according to previous work in CRL. The authors describe a specific learning framework based on mechanistic neural networks which encode the trajectories into not only the parameters of a flexible family of ODEs, but also hyperparameters of the solver, defining a flexible generative process of observed trajectories.

**Strengths:**

The authors state as motivation the need for successful real-world applications of CRL. This is an incredibly important problem to tackle, and cross-pollinating with the AI/ML for Science (specifically, inverse problems) community is a good place to start. There are many parallels between the areas of parameter identification in ODEs and latent variable identification in CRL: both recognize the significance of identifiability, but while it appears that application has outpaced theory in Neural ODEs, the opposite has occurred in CRL. The application of CRL theory to ODE inverse problems is hence original and potentially of great significance to both communities.

**Weaknesses:**

Although the paper draws mathematical connections between identifiability in inverse problems and CRL, I found the conceptual connection to be insufficient. The parameters in ODEs and latent variables in CRL, even if they can be made mathematically equivalent, are quite different in interpretation. The former represents physical parameters which need to be identified point-wise, whereas the focus in the latter (at least, within the scope of examples given in this paper) is typically in separating latent factors of variation, whereas identifiability of the factor itself can be up to arbitrary and unknown reparametrization (as long as it contains full and only information about that specific factor). For me, the lack of successful real world application of CRL is precisely because the community is unsure of what to do with these resulting factors, which have arbitrary units and are not physically interpretable.

To be fair, the usual ODE identifiability seems to be for the case where the dynamics are known, and not the "ODE discovery" setting that the paper focuses on in Sections 3.2+. Indeed, if all that is given are trajectories without an existing physical theory, it is clearly underdetermined to find physically meaningful parameters. Nonetheless, to build a truly useful bridge between the two fields, I find that the paper misses a crucial discussion on what this typical style of CRL identifiability, perhaps with unknown dynamics, can provide to the inverse problems community. For example, concerning the statement on l174:
> identifying climate zone-related parameters from sea surface temperature data could improve understanding of climate change because the impact of climate change significantly differs in polar and tropical regions.

The paper would greatly benefit by giving more details on why CRL-style identifiability is important in studying this problem. For example, does the ATE estimation (l377) rely on identifiability, and would non-identifiable contemporaries fail in causal prediction?

I really believe in the message of the paper and the mission to improve real-world applicability of CRL. However, unless the benefits of CRL-style identifiability for ODE inverse problems can be spelled out more explicitly, I do not believe the submission is ready for publication. Instead, I would strongly advocate for publication of a version where the following questions are at least partially answered.

**Questions:**

- (Repeated from above) Does the ATE estimation (l377) rely on identifiability, and would non-identifiable contemporaries fail in causal prediction?
- The experiments also show improvements in classification, generalization, and forecasting---is there an explanation, even an intuitive one, for why CRL-identifiable models in particular might be preferable for these tasks?
- Can the parameters learned by the pipeline always be interpreted as physical parameters of some underlying ground truth dynamics, or useful proxies thereof? Maybe if the MNN is well-specified?

### Other more minor points

- Is it straight-forward to port over CRL results that rely on restricting the function class of the generator, e.g., orthogonal/sparse Jacobian, when it is supposed to represent an ODE solver?
- The interpretation of dynamical systems as latent variable generative models may be the same approach taken in Bayesian inverse problems [1], for example see section 3.3 in the reference. If there is a connection here, maybe it should be discussed (disclaimer: I'm not at all an expert in Bayesian inverse problems).

[1] Stuart, Andrew M. "Inverse problems: a Bayesian perspective." Acta numerica 19 (2010).

**Limitations:**

As the paper does not introduce a new method but rather aims to draw connections, it is perhaps unnecessary to explicitly discuss the limitations. I think the authors do a fine job outlining potential future work in the final section.

---

> ### Author Rebuttal · Authors · 2024-08-05
>
> ### Reply to weaknesses
>
> We appreciate the reviewer’s thoughtful comments. We concur that identifying parameters from an unknown system is highly **challenging** and can be **difficult to interpret**. We also acknowledge that the **CRL-identifiability** provided in `Corollary 3.2` is **limited**, as it isolates certain parameters only up to a bijection. However, given that point estimates for parameter identification in **unknown** systems are **fundamentally impossible** (`lines 198-200`), our neural emulator, which achieves **strong forecasting** performance and provides **some level** of parameter identification, surpasses models that focus solely on forecasting without considering identifiability, as shown by comparison to the **non-identified TI-MNN** model in `Table 2` in the main text. We argue that CRL-identifiability remains valuable for various downstream analyses, even when the underlying system is unknown:
>
> 1. Identified latent parameters help **understand** **causal effects under intervention**. For example, in `Figure 6` (Appendix), the sea surface temperature drops when the inferred latitude-related variable $\\hat{\\theta}$ is permuted. This can be viewed as a **sensitivity analysis** within the dynamical system community, typically studied experimentally on a case-by-case basis.
> 2. Prior works [`19, 20, 21`] have demonstrated that **CRL-identified** latent variables **outperform** in related **downstream** classification tasks and exhibit **greater robustness** in **domain adaptation** **and out-of-distribution** scenarios. When these identifiable CRL algorithms are applied to **dynamical systems**, these advantageous properties are **retained**, as evidenced by `Figure 2` and `Table 2` in the main paper.
> 3. **Isolating covariates** information is crucial for treatment effect estimation, see the second point under `additional experiments` under [global rebuttal](https://openreview.net/forum?id=MWHRxKz4mq&noteId=n59QJ5gxJF).
>
> &nbsp;
>
> ### Reply to questions (same order as given by the reviewer)
>
> 1. Thank you for this question. ATE estimation **critically depends on identifiability**. Our additional experiments demonstrate that **non-identified** parameters yield **meaningless** ATE estimates, whereas **identified** parameters provide an **increasing** trend. We will update this figure in the revised manuscript to reflect these findings. Please refer to the **second point** in `additional experiments` under [global rebuttal](https://openreview.net/forum?id=MWHRxKz4mq&noteId=n59QJ5gxJF) for more details.
> 2. Intuitively, CRL-identified latent variables are valuable for downstream tasks because the learned representation accurately captures **all and only** the information about the ground truth parameters, **isolating confounding factors**, which is critical for treatment effect estimations (see `additional experiments`). For example, if a task is related to a parameter $\\theta$, a **non-identified** representation might include **arbitrary information unrelated** to $\\theta$. Using such representations for classification tasks could perform **no better than random guessing**. This is why we evaluate the learned representation on downstream classification tasks and robustness tests to validate its identifiability.
> 3. When the MNN (Mechanistic Neural Network [`22`]) is well-specified, the learned parameters can **correspond one-to-one** with physical parameters through a bijection. This bijection serves as a useful proxy in **sensitivity analysis** and provides **high-quality features** for downstream classification tasks, as previously discussed. If some physical parameters are known for certain trajectories, the **bijection** can be **learned explicitly**, allowing the mapping of the learned representation to the ground truth.
>
> &nbsp;
>
> ### Reply to other minor points (same order as given by the reviewer)
>
> 1. Thanks for raising this point. It is discussed in [global rebuttal](https://openreview.net/forum?id=MWHRxKz4mq&noteId=n59QJ5gxJF) under “Identifying ODEs with specific properties.”
> 2. We agree that interpreting dynamical systems as latent variable models parallels approaches used in Bayesian inverse problems. As discussed in `lines 158-164`, **most Bayesian methods for parameter identification**, such as Gradient Matching [`4`], assume a **known** functional form but often **lack clear identifiability statements**. In contrast, our `Corollary 3.1` provides a theoretical framework that **explains** the successful empirical identification results observed with these methods.

---

> > ### Author Response · Authors · 2024-08-12
> > **We kindly ask if the rebuttal has addressed the reviewer's concerns**
> >
> > Dear reviewer `aFgN`:
> >
> > We greatly appreciate the reviewer's constructive feedback and thoughtful consideration. As the discussion period concludes soon, we kindly ask if the rebuttal has addressed their concerns.
> >
> > Should any further questions or concerns arise, we are happy to provide clarification during the remaining discussion time.

---

> > > ### Comment · Reviewer_aFgN · 2024-08-12
> > >
> > > Dear Authors,
> > >
> > > Thank you for the thoughtful rebuttal. I appreciate the additional experiments and comments that show the benefits of identifiable CRL-style model in this setting. However, I am still not sure about the connection between the two identifiability theories. I think an unknown DE fundamentally changes the nature of the identifiability problem in a way that is not fully discussed by the authors. Despite this, I agree with you that many aspects of this problem (identifying parameters in unknown systems) is fundamentally impossible, yet discovering DEs is still of great importance. To this end, I think the types of regularization that CRL brings can be very interesting for future application.
> > >
> > > In view of this, I am willing to increase my score to "weak accept", assuming that the authors include elements of the global rebuttal into the paper, and thus further explaining to the DE community what the benefits and (theoretical) limitations of CRL are.

---

> ### Comment · Area_Chair_e8B7 · 2024-08-12
> **Author-Reviewer Discussion Reminder**
>
> Dear reviewers
>
> Since we are approaching the end of review-author discussion. Please read the author's response and clearly acknowledge that if they have successfully addressed your concerns.
>
> Best,
>
> You AC

---

### Official Review · Reviewer_3NaZ · 2024-07-12

**Soundness:** 3
**Presentation:** 3
**Contribution:** 3
**Rating:** 6
**Confidence:** 4

**Summary:**

This paper proposes a theory and methodology for representation learning from dynamical systems. In particular, it proposes a model in which latent causal variables deterministically generate an observed time-series trajectory through an ODE, with the task of identifying the latent variables from observed time series. Theoretically, the authors devise identifiability conditions by drawing the causal representation learning literature, treating the ODE as the (black box) mixing function. To identify variables in practice, it is proposed to use mechanistic neural networks (MNNs) to learn the ODE mixing function. Empirical results show that the proposed method can strike a good balance between identification of the latent variables and predictive performance.

**Strengths:**

The topic of the paper (learning interpretable causal variables from dynamical systems) is of great importance for e.g. the sciences and this paper makes a significant conceptual step in this direction. The specific proposed problem (recovery of stationary, trajectory specific parameters) and the idea of applying causal representation techniques (such as the multiview approach) to dynamical systems data is an interesting, and to my knowledge, novel problem and approach.

The proposed method for partially identifying parameters is a sensible combination of existing components (loss function from CRL literature, and MNN parameterization of ODE) that enables both accurate prediction of trajectories as well as identification of causal variables (by separating the causal variables $\theta$ from the ODE parameters $\alpha$).

The experiments are well-motivated and show the competitive/superior performance of the proposed method over baselines in terms of prediction and identification.

The paper is generally well written and has a clear structure, though some of the notation could be confusing at times (see suggestions below).

**Weaknesses:**

The novelty in terms of theory is fairly weak; the theoretical identifiability results directly follow from existing literature with minimal modifications, and does not exploit anything specific to ODEs (e.g. 1. are there certain ODE assumptions which enable identifiability, e.g. certain function types or sparsity in the ODE's equation; 2. how would the situation change with time-dependent latents)?

In terms of significance, the assumptions underlying the method are very strong and may limit the applicability of the method (e.g. determinism, no latent temporal variables).

**Questions:**

- The results show that one can identify the causal variables (up to blocks). Does this mean that the ODE function is also learned correctly, and thus we can trust e.g. the ATE in Experiment 6.2?
- What is forecast error in Table 2? (MSE?)

Suggestion on presentation: It can sometimes be confusing how $x, \theta, $ and the parameters in $\alpha$ relate. It might be useful to include a diagram (e.g. plate notation) to show which variables are time-dependent, trajectory-dependent etc.

**Limitations:**

The limitations of the method are mostly well-discussed in the text and conclusion. There are two other limitations I would encourage the authors to discuss. Firstly, as with most CRL methods, significant prior knowledge is required: in particular here, identifying twin trajectories where some unknown causal/latent parameters differ. Secondly, it is assumed that the ODE evolves over the observation space (e.g. in the sea surface temperature example, over the surface temperatures), whereas the true underlying ODE is more likely to involve other/latent temporal variables (e.g. amount of sunlight, polar ice cap coverage), on top of stationary latents.

---

> ### Author Rebuttal · Authors · 2024-08-05
>
> We thank the reviewer for the positive feedback and will address the concerns individually.
>
> &nbsp;
>
> ### Reply to weaknesses
>
> 1. Please refer to our general clarification on the **novelty and contribution** in [global rebuttal](https://openreview.net/forum?id=MWHRxKz4mq&noteId=n59QJ5gxJF).
>
> 2. > are there certain ODE assumptions which enable identifiability, e.g. certain function types or sparsity in the ODE's equation?
>
>     **Identification of ODEs with certain properties** is discussed in [global rebuttal](https://openreview.net/forum?id=MWHRxKz4mq&noteId=n59QJ5gxJF). Please refer to the paragraph  “Identifying ODEs with specific properties” for details.
>
> 3. > how would the situation change with time-dependent latents?
>
>    **Identification of time-varying latents** is discussed in `Appendix D`, and we will clarify this reference in the updated version, referencing it in the main text. This issue is closely related to the temporal CRL approaches that model dynamics in the latent space [`8, 16, 17`]. We offer a brief discussion of this in the [global rebuttal](https://openreview.net/forum?id=MWHRxKz4mq&noteId=n59QJ5gxJF) under "Why not model dynamics in the latent space"  and will address the topic more thoroughly in the revised manuscript.
> 4. The validity and applicability of our **technical assumptions** are discussed in the [global rebuttal](https://openreview.net/forum?id=MWHRxKz4mq&noteId=n59QJ5gxJF).
>
> &nbsp;
> ### Reply to questions
>
> 1. The ODE function is **learned implicitly** by the mechanistic neural network but with disentangled parameters, as evidenced by the **low forecasting error and OOD performance** (see `Table 2` in the main paper).
> 2. Thanks for this question. Yes, the forecasting error is the **mean squared error(MSE)** summed over the state dimension and averaged over batch and time dimensions. We will add this information in the updated manuscript.
>
> &nbsp;
> ### Reply to suggestion on the representation
>
> We thank the reviewer for this valuable advice. `Figure 6` in the Appendix illustrates such an overview of the architecture and relations between different variables (referred to in `lines 244-246`). We will clarify this link further in the revised version.

---

> > ### Author Response · Authors · 2024-08-12
> > **We kindly ask if the rebuttal has addressed the reviewer's concerns**
> >
> > Dear reviewer `3NaZ `:
> >
> > We greatly appreciate the reviewer's constructive feedback and thoughtful consideration. As the discussion period concludes soon, we kindly ask if the rebuttal has addressed their concerns.
> >
> > Should any further questions or concerns arise, we are happy to provide clarification during the remaining discussion time.

---

> > > ### Comment · Reviewer_3NaZ · 2024-08-12
> > >
> > > Thank you for the response. I will keep my positive rating.

---

> ### Comment · Area_Chair_e8B7 · 2024-08-12
> **Author-Reviewer Discussion Reminder**
>
> Dear reviewer
>
> Since we are approaching the end of review-author discussion. Please read the author's response and clearly acknowledge that if they have successfully addressed your concerns.
>
> Best,
>
> You AC

---

### Official Review · Reviewer_bh3V · 2024-07-13

**Soundness:** 3
**Presentation:** 3
**Contribution:** 3
**Rating:** 6
**Confidence:** 3

**Summary:**

This paper bridges causal representation learning (CRL) with dynamical system learning.

It introduces partially identifiable and practical models by merging methodologies from both CRL and dynamic systems.

The authors develop models capable of handling out-of-distribution classification tasks and treatment effect estimation.

Notably, they validate their approach using a wind simulator and real-world climate data, effectively demonstrating the model's potential in addressing causal questions related to climate change.

**Strengths:**

1. This paper is one of the few causal representation learning works that claim contributions to theoretically sound methods with working and impactful real-world problem solutions.

2. The authors provided a theoretically sound analysis of the partial identifiability of the model.

3. The task chosen by the authors to validate their proposed method is very interesting and potentially impactful: climate model estimation. Solving such a problem and providing solutions with theoretical guarantees will be very impactful.

**Weaknesses:**

The proposed method claims partial identifiability of the invariant part of the dynamic models. There are existing works, such as [1], which study dynamic models in hidden space and demonstrate partial identifiability for the invariant part of the dynamic models.

Can the authors provide some comments on why they chose to study the dynamic model in the observational space and how their proposed approach might be better than the existing work?



[1] Li, Zijian, et al. "When and How: Learning Identifiable Latent States for Nonstationary Time Series Forecasting." arXiv, 2024, arxiv.org/abs/2402.12767.

**Questions:**

Please see Weaknesses.

**Limitations:**

Yes.

---

> ### Author Rebuttal · Authors · 2024-08-05
>
> ### Reply to weaknesses
>
> Thank you for the positive feedback and for providing this interesting related work (listed as [`17`] under our references). Although our paper may **superficially resemble** [`17`], there are **important differences** between the two. Please allow us to make a quick clarification:
>
> 1. The **hierarchy of the problem setting** differs: At a high level, [`17`] models dynamics in the latent space by treating **time-varying** latent variables as hidden states, while we model dynamics directly in the **observational** space. A discussion on this distinction is provided in the [global rebuttal](https://openreview.net/forum?id=MWHRxKz4mq&noteId=n59QJ5gxJF) under "Why not model dynamics in the latent space."
> 2. The **mixing process** differs: We consider the **entire trajectory** as our entangled **observation**, generated by $(x\_{t\_1}, …, x\_{t\_N}) \= F(\\theta)$, where the **mixing** function $F$ is inherently **non-stationary** and depends on the timespan of interest. In contrast, the referred work [`17`] uses stationary diffeomorphism mixing and **implicitly** accounts for **non-stationary properties** within the latent space.
> 3. Our **definition of “stationary/invariant”** is different:  [`17`] considers a partition of latent variables $z\_t^s$ as stationary when it is **invariant to the environment** $e\_t$, meaning there is no causal link from the environment $e\_t$ to $z\_t^s$. However, $z\_t^s$ still depends on time. In contrast, in our context, a stationary parameter is **time-invariant** and trajectory-specific, such as the pole length $l$, gravity $g$, cart mass $m\_c$, and pole mass $m\_p$ in the cart pole experiment (see `additional experiments` under [global rebuttal](https://openreview.net/forum?id=MWHRxKz4mq&noteId=n59QJ5gxJF)).
> 4. Overall, we believe that [`17`] is more closely related to [`8,16`], and that both lines of work are **orthogonal yet equally important**. We will provide a thorough discussion of this connection in the updated manuscript.

---

> > ### Comment · Reviewer_bh3V · 2024-08-09
> >
> > Thank you. I've read your rebuttal, responses, and the other reviews. I will keep my score since it is already positive.

---

### Author Rebuttal · Authors · 2024-08-05

We are extremely grateful to the reviewers and AC for their time and valuable feedback. We very much appreciate that they found the problem we are tackling is *“very interesting and potentially impactful*”(`bh3V`), “*of great importance for e.g. science*”(`3NaZ`), “*incredibly important*”(`aFgN`), and “*intriguing and emerging*”(`yeqY`). We are happy to see that our idea of applying CRL techniques to dynamical systems is *“interesting and novel”* (`3NaZ`), “*original and potentially of great significance to both communities*”(`aFgN`).

**Novelty and contribution (`3NaZ`, `yeqY`)**
* As discussed in `lines 73-78`, this paper’s main contribution is providing **clear parameter-identifiability statements** for dynamical systems, whereas numerous **previous works** on ODE discovery [`1, 2, 3, 4`] refrained from doing so by explicitly stating that it is **unknown** which settings yield identifiability (`lines 165-166`).
* To the best of our knowledge, our paper is the **first** to target **real-world scientific applications** and successfully demonstrate identification results on **raw measurements**, unlike **previous CRL** work which relies on synthetic or **heavily pre-processed** data (e.g. manually rendered images[`5, 6`] or extracted avatar skeleton [`9`]).

**Validity of the technical assumptions (`3NaZ`, `yeqY`)**
* Assumptions 2.1 and 2.2 are **standard and necessary** for parameter identification in dynamical system [`10, 11`]. Many ODEs satisfy these assumptions, including Lotka-Volterra ODE, Van der Pol oscillator, and chaotic systems such as the Lorenz attractor and the Rössler attractor.
* Further, we justify in `Table 1` that **CRL assumptions** (`3.1-3.3`) **align** with standard assumptions 2.1 and 2.2, establishing the theoretical ground for cross-pollination between these two fields.
* Nevertheless, we acknowledge the limitation of the determinism assumption, as discussed in `Section 6`.

**Identifiability with ODE with specific properties (`3NaZ`, `aFgN`, `yeqY`)**
* `Corollary 3.1` provides full identifiability for stationary parameters in ODEs with known functional forms, including **any parametric form** (linear, polynomial, even nonlinear). This is **supported** by various empirical studies from **prior literature** on equation discovery [`1, 2, 3, 4`] and attached **additional experimental results**.
* For ODEs with partially known properties, such as a **sparse linear combination** $f_{\theta}(x, t) = \sum\_{i = 1}^{m} \theta_i \phi_i(x)$  **of various basis functions** $\phi_i$ within a comprehensive dictionary (e.g., SINDy-like scenarios [`1, 2, 3`]), `Corollary 3.1` ensures parameter identifiability under a sparsity constraint. Notably, the ODE relates to the parameters $\theta$ linearly, as detailed in the main paper (`lines 154-157`) and the appendix (`lines 653-663`).
* **Many existing CRL works** assume certain properties on the generating process, such as **sparsity** [`14`] or **specific functional class** [`15`], which can be **directly imported** into our framework by replacing the multiview approach.


**Why not model dynamics in the latent space (`bh3V,3NaZ,yeqY`)**
* We model dynamics directly in the observational space because the underlying parameters $\theta$ **directly determine** the whole trajectory: $(x_{t_1}, \dots, x_{t_N}) = F(\theta)$.
* **Almost all CRL works** assume the latent variables **directly influence** the observation, except for a few considering hierarchical latent models [`12, 13`], which we will thoroughly discuss in the updated manuscript.
* As discussed in `Section 6`, we agree assuming direct observation of the states is limiting, and relaxing this assumption is an interesting future direction. However, since this paper is the **first to formally connect** CRL-identifiability with inverse problems in dynamical systems, we believe that exploring this perspective is beyond the scope of the current study.


**Additional experiments (see attached pdf)**
* We provide additional experiment results on specific dynamical systems with **known functional forms**, including eight highly complex systems from **ODEBench**[`18`] and the **Cart-Pole** system inspired by [`8`].
  * For each system, we sample 100 tuples of parameters $\theta$ within a valid range to preserve properties like chaos. Each problem is solved exactly as described in `Corollary 3.1`.
  * The root-mean-square deviation (RMSE) is averaged across the parameter dimensions and sample size (100), reported via mean and std.
  * We observe **highly accurate point estimates** for all stationary system parameters $\theta$, validating `Corollary 3.1` **across various experimental settings**. Due to space constraints, we cannot enclose results for all 63 systems from ODEBbench[`18`]. Complete results and implementation details will be provided in the revised manuscript.
* Comparing **ATE** estimates from non-identified and identified representations for SST-V2:
  * The attached `Figure 1` illustrates the estimated ATE from the **non-identified** representation **lacks a discernible pattern**, while the **identified** one exhibits a noisy yet **clear increasing trend**, indicating the global warming effect.
  * This is because the non-identified representation failed to isolate the covariates $\theta$, leading to biased treatment effect estimates. To estimate treatment effects, the **covariates** (i.e., the latitude-related parameters we identify) **mustn’t be influenced by the treatment** (i.e., the climate zones). Otherwise, they become confounders, leading to incorrect estimates [`22`].
  * We apologize for not explaining this clearly in the current draft. In the final version, we will add the additional graphical model and the non-identified baseline (attached `Figure 1`) together with a thorough explanation about treatment effect estimation with covariates and why CRL-identifiability (up to bijection) is necessary to avoid confounding.

---

### Author Response · Authors · 2024-08-05
**References**

[1] Steven L Brunton, et al. Discovering governing equations from data by sparse identification of nonlinear dynamical systems. Proceedings of the national academy of sciences, 113(15):3932–3937, 2016\.

[2] Steven L Brunton, et al. Sparse identification of nonlinear dynamics with control (sindyc). IFAC-PapersOnLine, 49(18):710–715, 2016\.

[3] Alan A Kaptanoglu, et al. Promoting global stability in data-driven models of quadratic nonlinear dynamics. Physical Review Fluids, 6(9):094401, 2021\.

[4] Philippe Wenk, et al. Fast gaussian process based gradient matching for parameter identification in systems of nonlinear odes. In The 22nd International Conference on Artificial Intelligence and Statistics, pages 1351–1360. PMLR, 2019\.

[5] Julius Von Kügelgen, et al. Self-supervised learning with data augmentations provably isolates content from style. Advances in neural information processing systems, 34: 16451–16467, 2021\.

[6] Dingling Yao, et al. Multi-view causal representation learning with partial observability. In The Twelfth International Conference on Learning Representations, 2024\.

[7] Klindt, David, et al. "Towards nonlinear disentanglement in natural data with temporal sparse coding." arXiv preprint arXiv:2007.10930 (2020).

[8] Yao, Weiran, et al. "Temporally disentangled representation learning." Advances in Neural Information Processing Systems 35 (2022): 26492-26503. (referred to as [a] by reviewer `yeqY`)

[9] Balsells-Rodas, Carles, et al. "On the Identifiability of Switching Dynamical Systems." arXiv preprint arXiv:2305.15925 (2023).

[10] Ernest Lindelöf. Sur l’application de la méthode des approximations successives aux équations différentielles ordinaires du premier ordre. Comptes rendus hebdomadaires des séances de l’Académie des sciences, 116(3):454–457, 1894\.

[11] Ror Bellman and Karl Johan Åström. On structural identifiability. Mathematical biosciences, 7 (3-4):329–339, 1970\.

[12] Kong, Lingjing, et al. "Identification of nonlinear latent hierarchical models." Advances in Neural Information Processing Systems 36 (2023): 2010-2032.

[13] Xie, Feng, et al. "Identification of linear non-gaussian latent hierarchical structure." International Conference on Machine Learning. PMLR, 2022\.

[14] Danru Xu, et al. A sparsity principle for partially observable causal representation learning. International Conference on Machine Learning, 2024\.

[15] Ahuja, Kartik, et al. "Interventional causal representation learning." International conference on machine learning. PMLR, 2023\.

[16] Song, Xiangchen, et al. "Temporally disentangled representation learning under unknown nonstationarity." Advances in Neural Information Processing Systems 36 (2024).

[17] Li, Zijian, et al. "When and How: Learning Identifiable Latent States for Nonstationary Time Series Forecasting." arXiv preprint arXiv:2402.12767 (2024). (referred to as [1] by reviewer `bh3V`)

[18] d'Ascoli, Stéphane, et al. "Odeformer: Symbolic regression of dynamical systems with transformers.” In International Conference on Learning Representations, 2024\.

[19] Locatello, Francesco, et al. "Weakly-supervised disentanglement without compromises." International conference on machine learning. PMLR, 2020\.

[20] Fumero, Marco, et al. "Leveraging sparse and shared feature activations for disentangled representation learning." Advances in Neural Information Processing Systems 36 (2024).

[21] Lachapelle, Sébastien, et al. "Synergies between Disentanglement and Sparsity: Generalization and Identifiability in Multi-Task Learning." arXiv preprint arXiv:2211.14666 (2022).

[22] Adeel Pervez, et al. Mechanistic neural networks for scientific machine learning. International Conference on Machine Learning, 2024\.

[22] Feuerriegel, Stefan, et al. "Causal machine learning for predicting treatment outcomes." Nature Medicine 30.4 (2024): 958-968.

---

### Author Response · Authors · 2024-08-12
**Thanks to all reviewers and the AC**

We would like to express our sincere gratitude to all the reviewers for their constructive feedback. The reviews and discussions have significantly enhanced the quality of our paper, and we will incorporate all suggested changes in the updated manuscript.

We also extend our thanks to the area chair and the reviewers for their time and effort. We greatly appreciate your engagement and will acknowledge your anonymous contribution in the revised paper.

---

### Decision · Program_Chairs · 2024-09-25

**Decision:**

Accept (poster)

**Comment:**

This paper proposes a connection between causal representation learning and dynamic system, ODE, to establish the identifiability of the dynamics system. Reviewers recognise the contribution of the paper, although the technical novelty is limited. During the rebuttal period, the author's responses addresses many concerns from the reviewers, so it is recommended to include these in the revised version.